# Cancer-associated USP28 missense mutations disrupt 53BP1 interaction and p53 stabilization

Hazrat Belal 🄳, Esther Feng Ying Ng 🄳, Midori Ohta 🄳 & Franz Meitinger 🄳 ✉

Cellular stress response pathways are essential for genome stability and are frequently dysregulated in cancer. Following mitotic stress, the ubiquitin-specific protease 28 (USP28) and the p53-binding protein 1 (53BP1) form a stable, heritable complex to stabilize the tumor suppressor p53, triggering cell cycle arrest or apoptosis. Here, we demonstrate that USP28 stabilizes p53 through deubiquitination. We further show that USP28 is required not only for an efficient stress response but also for maintaining basal p53 levels in some cancer cells. Loss of functional USP28 allows cells to evade mitotic stress and DNA damage responses in a manner that is specific to cell type and cancer context. We identify a prevalent, shorter USP28 isoform critical for p53 stabilization. Its C-terminal domain mediates PLK1-dependent binding to 53BP1, a dimerization-driven interaction necessary for mitotic stress memory, p53 stabilization, and cell cycle arrest. Cancer-associated missense mutations in this domain disrupt 53BP1 binding, impair nuclear localization, and destabilize USP28, compromising p53 stabilization. Notably, mutations in the 53BP1-binding domain occur more frequently in tumors than those in the catalytic domain, suggesting a potential role in cancer progression and implications for therapeutic strategies.

Mechanisms that regulate cell proliferation and genome stability are critical for tissue homeostasis. In response to cellular stress, diverse pathways converge on the tumor suppressor p53, leading to its stabilization and activation. This triggers cell cycle arrest or apoptosis—key barriers against genome instability and oncogenic transformation. The ubiquitin-specific protease 28 (USP28) has emerged as a key player in this process. While USP28 is known to promote tumor growth by stabilizing MYC in certain cancers[1–5], it has also been implicated in tumor suppression through p53 stabilization under stress conditions[6–10]. These seemingly paradoxical roles raise important questions about the context-specific functions of USP28 in cancer. Notably, studies linking USP28 to MYC stabilization have primarily focused on p53-deficient cancer cells, suggesting that USP28's role may shift depending on p53 status. Here, we define the molecular mechanism that engages USP28 in stress responses across normal and p53-wildtype cancer cells.

Systematic analysis of the cancer-associated mutations have identified USP28 and the p53-binding protein 1 (53BP1) as tumor suppressors, though the underlying mechanisms remain unclear[11]. In 2016, three independent studies revealed that both proteins detect mitotic stress-induced prolonged mitosis, stabilizing p53 to trigger cell cycle arrest or apoptosis[7–9]. This pathway, termed the "mitotic stopwatch" or "mitotic surveillance" pathway, protects cells from mitotic stress that could lead to chromosome missegregation and genome instability—hallmarks of cancer[6–9,12–14]. While both USP28 and 53BP1 have been observed at DNA damage sites, USP28 does not directly participate in DNA repair but may contribute to local p53 activation[10,15–17]. Notably, USP28's roles in mitotic stress response and DNA damage response appear to be independent[7–9,13,15,18,19].

Recent work has shed light on the molecular basis of the mitotic stopwatch pathway. During prolonged mitosis, the mitotic kinase PLK1 facilitates the assembly of a USP28–53BP1–p53 complex[13]. 53BP1 acts as

Okinawa Institute of Science and Technology Graduate University, Okinawa, Japan. ✉e-mail: franz.meitinger@oist.jp

a scaffold, recruiting USP28 and p53 via distinct domains, while PLK1-mediated phosphorylation of 53BP1 promotes p53 binding. However, the mechanisms regulating USP28's engagement in this process remain unclear. After mitotic stress, this complex persists in daughter cells, leading to p53 accumulation. Although p53 accumulation depends on USP28, the underlying molecular mechanism is still unclear[7,8]. In G1 phase, p53 induces the expression of p21[CDKN1A], inhibiting CDK4/6 to prevent cell cycle progression. Alternatively, p53 can trigger apoptosis, as observed in human embryonic stem cells and mouse embryo development[13,20].

Growing evidence highlights the mitotic stopwatch's role in maintaining tissue integrity during development[19–26]. Centrosomal defects, which prolong mitosis in the developing mouse embryo, have been shown to induce p53-mediated apoptosis[20]. Mutations in centrosomal genes frequently cause primary microcephaly, a condition associated with excessive mitotic stress in neuronal progenitors[19,27]. Deletion of USP28 or 53BP1 alleviates these developmental defects, underscoring their critical role in responding to mitotic stress. Similar effects have been observed in epidermal stratification and embryonic lung and kidney development, further supporting the importance of USP28 in safeguarding tissue integrity[20,22,24–26]. Considering the importance of USP28, it is crucial to understand the molecular modules required for the implementation of mitotic stress response.

Structurally, USP28 is a unique ubiquitin-specific protease (USP) containing an internal dimerization arm[28,29]. Its USP domain is essential for both its tumor-suppressive and oncogenic functions[5,7]. Two N-terminal ubiquitin-binding domains (UIM, UBA) have been proposed to mediate substrate interactions, though their role in substrate specificity remains unclear[30]. USP28 dimerization is critical for its function[28,29], but recent findings suggest that DNA damage induces ATM-dependent phosphorylation, locking USP28 in a monomeric state that promotes MYC stabilization and genome instability, potentially contributing to tumorigenesis by counteracting SCF[FBW7]-mediated MYC degradation[3]. The function of the C-terminal region, which comprises nearly 40% of the protein, remains largely unknown. How USP28 coordinates its tumor-suppressive and oncogenic roles is a fundamental open question

In this study, we elucidate the molecular mechanism by which USP28 regulates p53 stability. We demonstrate that cancer-associated mutations in the C-terminal region of USP28 disrupt its dimerization- and PLK1-dependent interaction with 53BP1, selectively impairing its ability to stabilize p53 and coordinate stress responses. Notably, we find that a subset of cancer cells relies on USP28 to maintain basal p53 levels. As a result, USP28-deficient cancer cells exhibit attenuated responses not only to mitotic stress but also to DNA damage. Thus, USP28 missense mutations enable continued proliferation under stress conditions, potentially promoting genomic instability and driving tumor progression.

## Results

### USP28 deubiquitinates and stabilizes p53

USP28 has been shown to stabilize the oncogene MYC to drive cell proliferation in response to DNA damage and the tumor suppressor p53 to cease cell proliferation in response to mitotic stress-induced prolonged mitosis or DNA damage[6–9,13,17,18]. Considering these reported opposing functions of USP28, we asked the question whether USP28 knockout confers an advantage or disadvantage to cancer cells that experience mitotic stress or DNA damage, specifically in untransformed and cancer-derived p53-wildtype cell lines. To do so, we developed a cell proliferation competition assay (Fig. 1A). A mixture of wildtype and USP28Δ cells was treated with the anti-mitotic inhibitor for PLK4 (PLK4i, centrinone, 150 nM) or the DNA damaging compound Doxorubicin (DXR, 10 nM). PLK4i induces mitotic stress by prolonging mitosis for 60–150 min and DXR introduces DNA strand breaks[9,31]. The

concentrations of both drugs were titrated to obtain similar effects on cell proliferation in p53-wildtype and p53-depleted hTERT RPE-1 (RPE1) cells (Fig. 1B). After eight days of treatment, the percentage of knockout cells was compared to wildtype cells. An enrichment of USP28Δ cells suggests that p53-dependent cell cycle arrest is the dominant mechanism, whereas a decrease indicates that MYC-dependent promotion of cell cycle progression prevails. We also performed this assay for 53BP1, which is required for USP28-dependent p53 stabilization. For this experiment, we selected one non-transformed and eleven cancer cell lines that express wildtype p53 and exhibit comparable proliferation rates. We found that the deletion of USP28 or TP53BP1 (53BP1) significantly reduced the sensitivity of several cancer cells to PLK4i treatment, mirroring the response observed in untransformed RPE1 cells (Fig. 1C). In contrast, loss of USP28 resulted in only a modest change in sensitivity to DXR-induced DNA damage, whereas deletion of TP53BP1 had no detectable effect. The difference in effects between loss of USP28 and TP53BP1 could be explained by two potential models: (i) the absence of 53BP1-mediated DNA repair may increase sensitivity to DXR, which could counterbalance any protective effects normally provided by 53BP1; or (ii) the reduced sensitivity to DXR observed upon USP28 loss may involve a 53BP1-independent function of USP28 in the regulation of DNA damage responses[32].

To further investigate the role of USP28 in mitotic stress and DNA damage responses, we analyzed its impact on the stability of p53 and MYC. We selected two cell lines for this analysis: RPE1 cells, where USP28 loss attenuates the response to mitotic stress but not DNA damage, and A549 cells, where USP28 loss attenuates responses to both stressors. Both mitotic stress and DNA damage induced p53 stabilization, increased p21 expression, and downregulated MYC in both cell lines (Fig. 1D). The p53 response to PLK4 inhibition was USP28-dependent in both RPE1 and A549 cells, whereas the p53 response to DXR was USP28-dependent only in A549 cells. RPE1 cells harboring the 53BP1 G1560K mutation, which disrupts USP28 binding, phenocopied USP28 deletion, suggesting that these effects rely on the USP28–53BP1 interaction[13,17]. Surprisingly, we found that USP28 deletion lowered basal p53 levels in unstressed cancer cells (A549 and U2OS) (Figs. 1D and S1A). Probably due to the reduced p53 levels, USP28Δ A549 cells failed to respond to both mitotic stress and DNA damage (Fig. 1D). In contrast to previous studies[3–5], USP28 deletion had no effect on the half-life of MYC in the four tested p53 wildtype cell lines (Fig. S1B–E).

To investigate how mitotic stress increases p53 levels, we treated RPE1 cells with cycloheximide (CHX) to block translation. p53 levels declined during CHX treatment, indicating that p53 abundance is regulated mainly through protein degradation rather than transcription or translation (Fig. 1E, F). Basal p53 half-life was 11 minutes in RPE1 WT cells and 21 minutes in USP28Δ cells. After 4 days of PLK4 inhibition to induce mitotic stress, p53 half-life increased to 56 minutes in RPE1 WT cells but remained unchanged in USP28Δ cells. These results indicate that USP28 stabilizes p53 specifically in response to mitotic stress.

If USP28 stabilizes p53 through mitotic stress-dependent deubiquitination, USP28Δ cells should show an increased ratio of ubiquitinated to deubiquitinated p53. To test this, we immunoprecipitated p53 under denaturing conditions following PLK4i treatment (Fig. S1F). We observed a marked increase in ubiquitinated p53 in USP28Δ cells, supporting the model that USP28 prevents ubiquitin-mediated degradation of p53 (Fig. 1G)[7].

In summary, these findings suggest that USP28 mediates the mitotic stress response by deubiquitinating and stabilizing p53. While in RPE1 cells USP28 activity toward p53 is specific to mitotic stress, in certain cancer cells, USP28 is essential for maintaining baseline p53 levels even under unstressed conditions. Consequently, USP28 loss disrupts both mitotic stress and DNA damage responses.

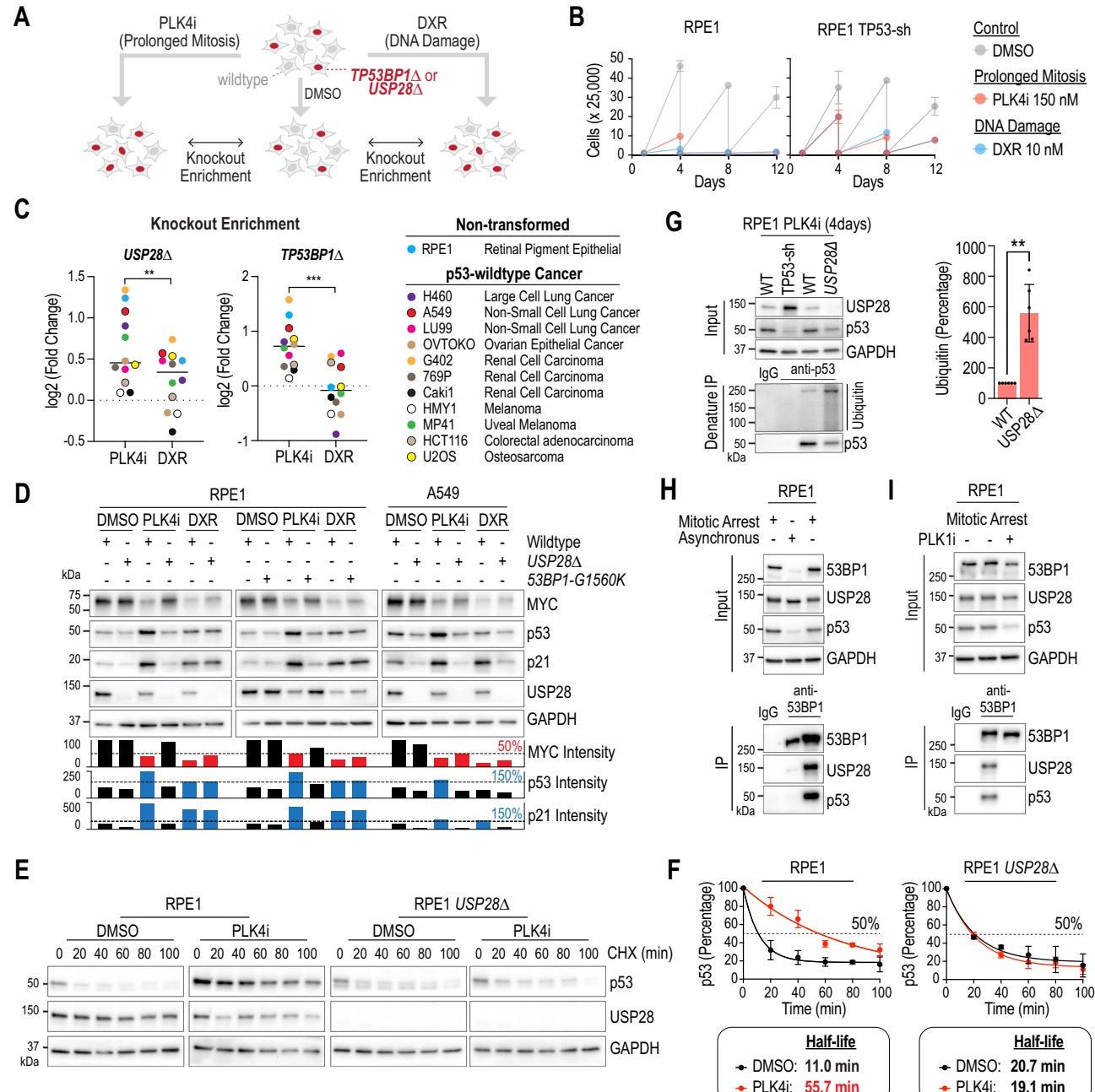

**Fig. 1 | USP28 deubiquitinates and stabilizes p53. A** Schematic of the competition assay used to assess USP28- and 53BP1-dependent drug sensitivity. Wildtype and knockout cells were mixed and treated with PLK4 inhibitor (PLK4i, 150 nM) or Doxorubicin (DXR, 10 nM). After 8 days, knockout cell abundance was quantified by sequencing; enrichment indicates gene-dependent drug sensitivity. **B** Proliferation of RPE1 and RPE1 TP53-sh cells treated with PLK4i or DXR. Mean ± SD from triplicates. Representative of two independent repeats. **C** Enrichment of *USP28Δ* and *TP53BP1Δ* cells after 8 days treatment. Two-sided Student's t-test (USP28: **P = 0.0021 with CI (95%) = −0.5640 to −0.1641; 53BP1: ***P = 0.0002 with CI (95%) = −1.115 to −0.4623). Each datapoint represents the mean of two independent experiments and reflects the differential abundance of the gene-deleted population, determined by pooled Sanger deconvolution. **D** Immunoblot of RPE1 and A549 cells with indicated genotypes after 4 days of treatment. Blue and red bars mark samples with p53/p21 enrichment (>1.5× DMSO) and MYC down-regulation (<0.5× DMSO), respectively. GAPDH, loading control. Samples derive from the same experiment but different gels for USP28, p53, GAPDH, p21, and

another for MYC were processed in parallel. Representative of two independent experiments. **E** Cycloheximide (CHX) chase assay assessing p53 half-life in RPE1 WT and *USP28Δ* cells treated with DMSO or PLK4i for 4 days. GAPDH, loading control. Representative of three independent repeats. **F** Quantification of CHX chase assays. Protein decay fitted exponential curve. Half-life = time to 50% loss. Mean ± SD. **G** Ubiquitination assay. Inputs, soluble supernatants. IP, immunoprecipitates. GAPDH, loading control. The samples derive from the same experiment but different gels for USP28, p53, GAPDH, and another for Ubiquitin were processed in parallel. Representative of six independent experiments. Data represent mean ± SD. Two-sided Student's t-test (**P = 0.0011 with CI (95%) = 254.4−656.4). **H, I** 53BP1 immunoprecipitation showing complex formation with USP28 and p53 in asynchronous and mitotically arrested cells (Nocodazole, 100 ng/ml, 16 h), without (**H**) or with (**I**) PLK1 inhibition (PLK1i, 100 nM). Input/IP/GAPDH as above. Representative of four (**H**) and three (**I**) independent experiments. Source data are provided as a Source Data file.

### USP28 isoform-specific response to mitotic stress

Our work raises the question of what activates USP28 toward p53 in mitotically stressed cells. A previous study has shown that mitotic stress leads to the formation of a mitotic stopwatch complex comprising USP28, 53BP1, and p53[13] (Fig. 1H). This complex forms specifically during prolonged mitosis, as evidenced by its dependence on the kinase PLK1 (Fig. 1I).

To recapitulate the mechanism of USP28-dependent p53 stabilization, we established a system in RPE1 cells where we can replace wildtype USP28 with mutant transgenes. We first assessed the USP28-dependent sensitivity of RPE1 cells to mitotic stress (extended mitosis) using a live-cell imaging approach (Figs. 2A and S1G)[9,13]. Cells were labeled with H2B-RFP and transiently treated with the KIF11^Eg5 inhibitor Monastrol, which induces a reversible mitotic arrest[18]. During treatment, cells were monitored and tracked for six hours using fluorescence imaging. Each cell in the asynchronous population entered mitosis at different time points and remained arrested in mitosis. Following Monastrol washout, cells exited mitosis and completed cell division. This approach gave rise to daughter cells whose mother cells had different mitotic lengths. Daughter cells were then tracked for an additional 48 hours to assess their fate.

As previously reported, the progeny of cells that spent over ninety minutes in mitosis exhibited a stable cell cycle arrest (Fig. 2A)[9,13,18]. However, the deletion of USP28 rendered cells less sensitive to mitotic stress, allowing them to continue proliferating despite a parental mitotic duration exceeding ninety minutes (Fig. 2A). This sensitivity to mitotic stress has been shown to be dependent on the formation of a complex involving 53BP1, USP28, and p53 (Fig. 1H)[13]. We observed that this complex remains stable following mitotic exit, facilitating p53 stabilization and subsequent p21 expression in G1 phase (Fig. 2B). In USP28-deleted cells, the response to prolonged mitosis was impaired, resulting in a failure to stabilize p53 and p21 (Fig. 2B).

The canonical isoform USP28^hIF1 (human isoform 1, NP_065937.1, NM_020886.4) encodes a 1077-amino acid protein (Figs. 2C and S2A). To investigate its role, we generated a USP28 knockout cell line ectopically expressing USP28^hIF1 but found that the longer isoform USP28^hIF1 did not interact with 53BP1 in mitotically arrested cells (Fig. 2D). In contrast, the shorter isoform, USP28^hIF2 (human isoform 2, NP_001333187.1, NM_001346258.2), successfully interacted with 53BP1, suggesting that isoform 2, rather than isoform 1, mediates the mitotic stress response. Notably, isoform 2 lacks exon 19, which encodes a 32-amino acid sequence in the C-terminal domain (Figs. 2C and S2A). Structural predictions from AlphaFold with high-confidence modeling indicate that exon 19 forms an additional alpha helix in isoform 1 (Figs. 2E and S2B, C). Our data suggests that this additional alpha helix obstructs the interaction surface for 53BP1 (Fig. 2C, D). We established single clones that express similar amounts of USP28^hIF1 and USP28^hIF2 as observed in parental RPE1 cells (Fig. S2D). In line with the interaction capability, only USP28^hIF2 but not USP28^hIF1 expressing cells induced cell arrest following prolonged mitosis (Fig. 2F).

Prompted by these findings, we analyzed the expression levels of both isoforms in RPE1 cells and found that USP28^hIF2 is predominantly expressed (Figs. 2G and S2E). To explore whether isoform-specific expression is consistent across different cell types, we extended our analysis to 22 p53-wildtype cancer cell lines from 10 tissue origins (Fig. S2F). In all cell lines examined, the short isoform USP28^hIF2 was the predominant transcript (Fig. 2H). We also analyzed the expression of USP28 isoforms in six mouse tissues, where the shorter isoform is designated as isoform 1 (USP28^mIF1, NP_780691.2, NM_175482.3), which exhibits similarity to human isoform 2 and the longer isoform is designated as isoform 2 (USP28^mIF2, NP_001346668.1, NP_780691.2), resembling human isoform 1 (Fig. S2A). In most mouse tissues, the shorter isoform was also dominant, except in the brain, where a higher proportion of isoforms containing exon 19 was detected (Fig. 2I). Sequencing confirmed that this band corresponded to exon 19 of the

longer isoform of USP28 (Fig. S2G). We subsequently tested additional human and mouse cell lines originating from neuronal tissues; however, none expressed notable levels of the longer isoform (Fig. S2H, I).

In conclusion, our findings demonstrate that the short isoform USP28^hIF2 is essential for the mitotic stress response, while the long isoform USP28^hIF1 is unable to fulfill this role. This suggests an isoform-specific regulatory mechanism for USP28 in the mitotic stress response and highlights the importance of the USP28 C-terminus.

### Identification of USP28 domains that are required for p53 stabilization

The USP domain of USP28 is required for p53 stabilization[7]. However, the specific motifs and regions of USP28 necessary for 53BP1 interaction and mitotic stopwatch function remain unknown. Thus, we set out to determine the functions of the regions that are N-terminal and C-terminal of the USP domain of USP28. To do so we took advantage from a surprising observation. When we expressed USP28^hIF2 in USP28Δ RPE1 cells, we noted that 16 out of the 34 isolated cell clones carried mutations in USP28^hIF2 (Figs. 3A, B and S3A). Each mutation was unique to a specific clone, ruling out the possibility of mutations originating from the lentiviral construct used to express the transgene. Five clones contained frameshift mutations, and two carried nonsense mutations with premature stop codons. Since USP28 overexpression is toxic in RPE1 cells[7], we reasoned that all the observed mutations specifically impair p53 stabilization (Figs. 3B and S3A). The identified missense mutations were located in the ubiquitin-specific protease domain (USP; aa149–399 and aa480–650; 5 clones), the dimerization arm (DA; aa400–579; 3 clones), and the C-terminus (aa651–1045; 5 clones). We selected six clones with one or two mutations in USP28^hIF2 and assessed their expression levels. For comparison, we also generated a clone expressing a catalytically inactive mutant of USP28^hIF2 (C171A)[7]. All tested USP28^hIF2 mutants were expressed at levels similar to or higher than endogenous USP28 (Fig. S3B).

To assess the functionality of the USP28^hIF2 mutants, cells were labeled with H2B-RFP and analyzed for sensitivity to prolonged mitosis using live imaging and single-cell tracking (Figs. 2A and 3C). In wildtype RPE1 cells, 94% of cells that experienced mitotic durations exceeding 90 minutes underwent arrest. In contrast, the C171A mutant and the USP28^hIF2 mutants displayed significantly reduced sensitivity to prolonged mitosis. Among the mutants, only 13% to 33% of cells were arrested following extended mitosis (>90 minutes), indicating that these mutations impair USP28's role in the mitotic stress response (Fig. 3C).

While clones with mutations in the ubiquitin-specific protease domain (C171A; D255N P287S; H600Y) and one clone with a frameshift mutation in the C-terminus (clone 16, D1003fs) exhibited nuclear localization similar to wildtype USP28^hIF2, two clones with mutations in the C-terminus (G903R and P953L) failed to localize to the nucleus (Figs. 3D and S3C). USP28^hIF2 has a predicted nuclear localization sequence (NLS) in the N-terminus (aa135–145; Figs. 3B and S3A), but this sequence does not explain the altered cytoplasmic localization observed in the G903R and P953L mutants (Figs. 3B, D and S3A, C). The expression levels of all tested mutants, except G903R and P953L, were 5- to 20-fold higher than in wildtype RPE1 cells (Figs. 3D and S3B, C). To assess p53 activation, we treated cells with the PLK4 inhibitor to prolong mitosis and measured stabilized p53 in the nucleus three days after the start of the treatment (Fig. 3D). We found that all mutant clones, even when overexpressed, failed to sufficiently stabilize p53 after prolonged mitosis (Fig. 3D).

To investigate the molecular defects in the USP28^hIF2 mutants, we assessed their ability to form a complex with 53BP1 during prolonged mitosis (Fig. 3E, F). Mutations in the USP domain (C171A; D255N P287S) did not affect USP28^hIF2 binding to 53BP1 (Fig. 3E). However, after release from mitotic arrest into the G1 phase, the complex remained stable but failed to stabilize p53 in the tested USP mutants (Fig. 3E).

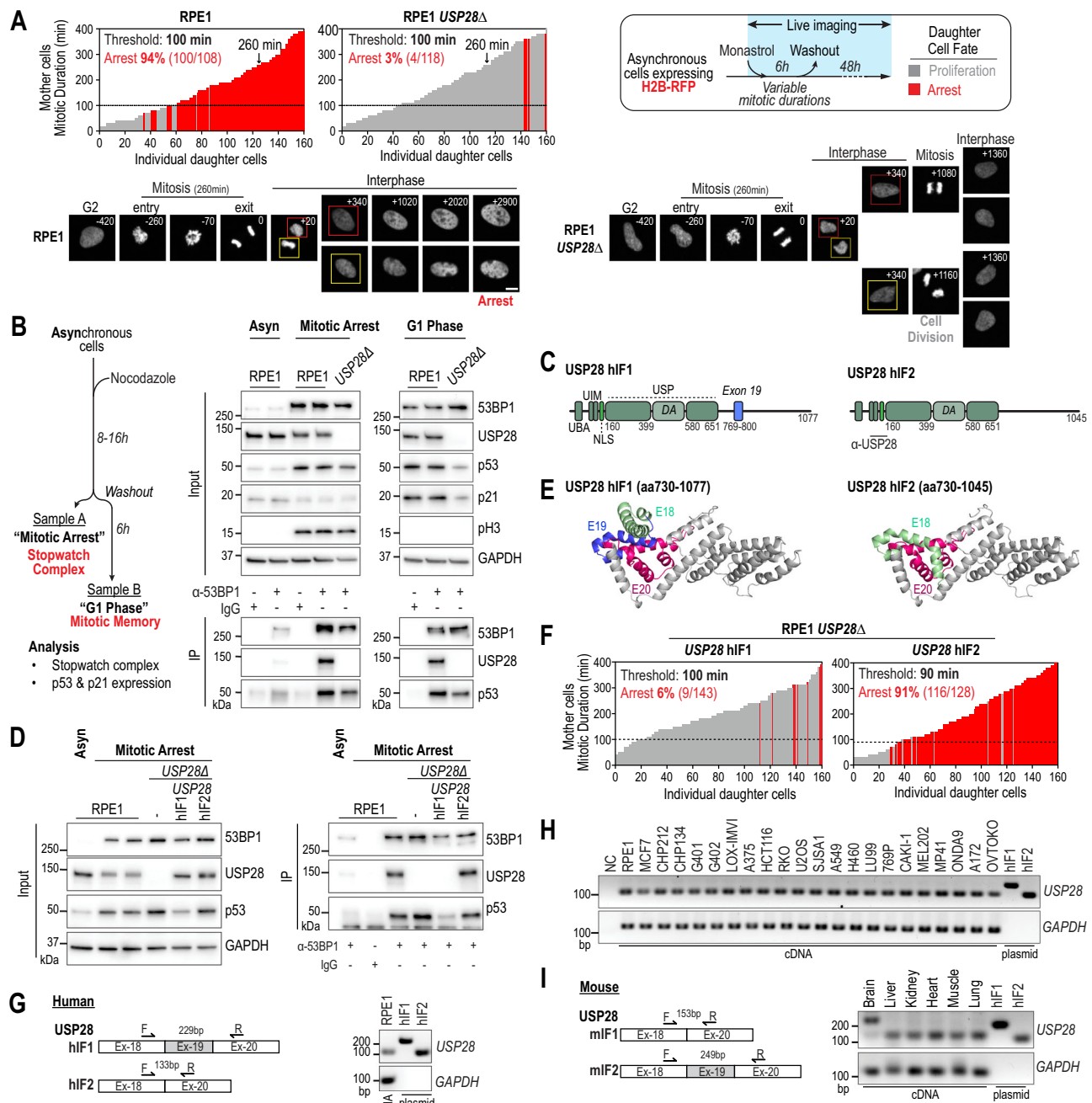

**Fig. 2 | Isoform-specific requirement of USP28 for mitotic stress-induced stabilization of p53. A** Imaging-based assay evaluating daughter cell fate after prolonged mitosis in WT and *USP28Δ* RPE1 cells. Cells were transiently arrested in mitosis using Monastrol, washout out, and imaged for 48 h. Each bar represents a daughter cell (grey, division; red, arrest); bar height reflects mother cell mitotic duration. The arrest threshold marks the mitotic duration at which >50% of daughters arrest. Representative examples shown (260 min mitotic duration). n = 160 cells per graph, pooled from 3 independent replicates (RPE1: 80, 58, 22; RPE1 *USP28Δ*: 54, 39, 67). Scale bar: 10 μm. **B** 53BP1 immunoprecipitation from WT and *USP28Δ* RPE1 cells arrested in mitosis (Nocodazole, 8–16 h) or released into G1. "Mitotic Arrest" samples detect stopwatch complexes (53BP1–USP28–p53); "G1 Phase" samples assess p53 activation. IP, immunoprecipitates. Inputs, soluble supernatants. GAPDH, loading control. pH3, mitosis marker. The input samples derive from the same experiment but different gels for 53BP1, USP28, p53, GAPDH, p21, and another for pH3 were processed in parallel. Representative of two independent experiments. **C** Schematic of human USP28 isoforms. Exon 19 is present in

*USP28*[hIF1] but not *USP28*[hIF2]. hIF1, human isoform 1; hIF2, human isoform 2; UBA, Ubiquitin-associated domain; UIM, Ubiquitin-interacting motif; NLS, Nuclear localization sequence; USP, Ubiquitin-specific protease; DA, Dimerization arm. Antibody epitope (α-USP28) indicated. **D** 53BP1 immunoprecipitation in *USP28Δ* cells expressing USP28[hIF1] or USP28[hIF2] (Fig. 2B). IP, immunoprecipitates. Inputs, soluble supernatants. GAPDH, loading control. Representative of two independent experiments. **E** AlphaFold model of USP28 C-terminus showing isoform-specific differences. Exons 18- 20 indicated (E18-E20). **F** Imaging assay as in (**A**), showing USP28[hIF2]-expressing cells arrest after prolonged mitosis, unlike hIF1-expressing cells. n = 160 cells per graph, pooled from 3 independent biological replicates (*USP28* hIF1: 76, 67, 17; *USP28* hIF2: 61, 50, 59). **G–I** RT-PCR analysis of exon 19 expression. Exon 19 is not detected in RPE1 cells (**G**), across 21 cancer lines (**H**), or mouse tissues other than the brain (**I**). Controls: plasmids for hIF1 and hIF2. GAPDH, loading control. Representative of two independent experiments. Source data are provided as a Source Data file.

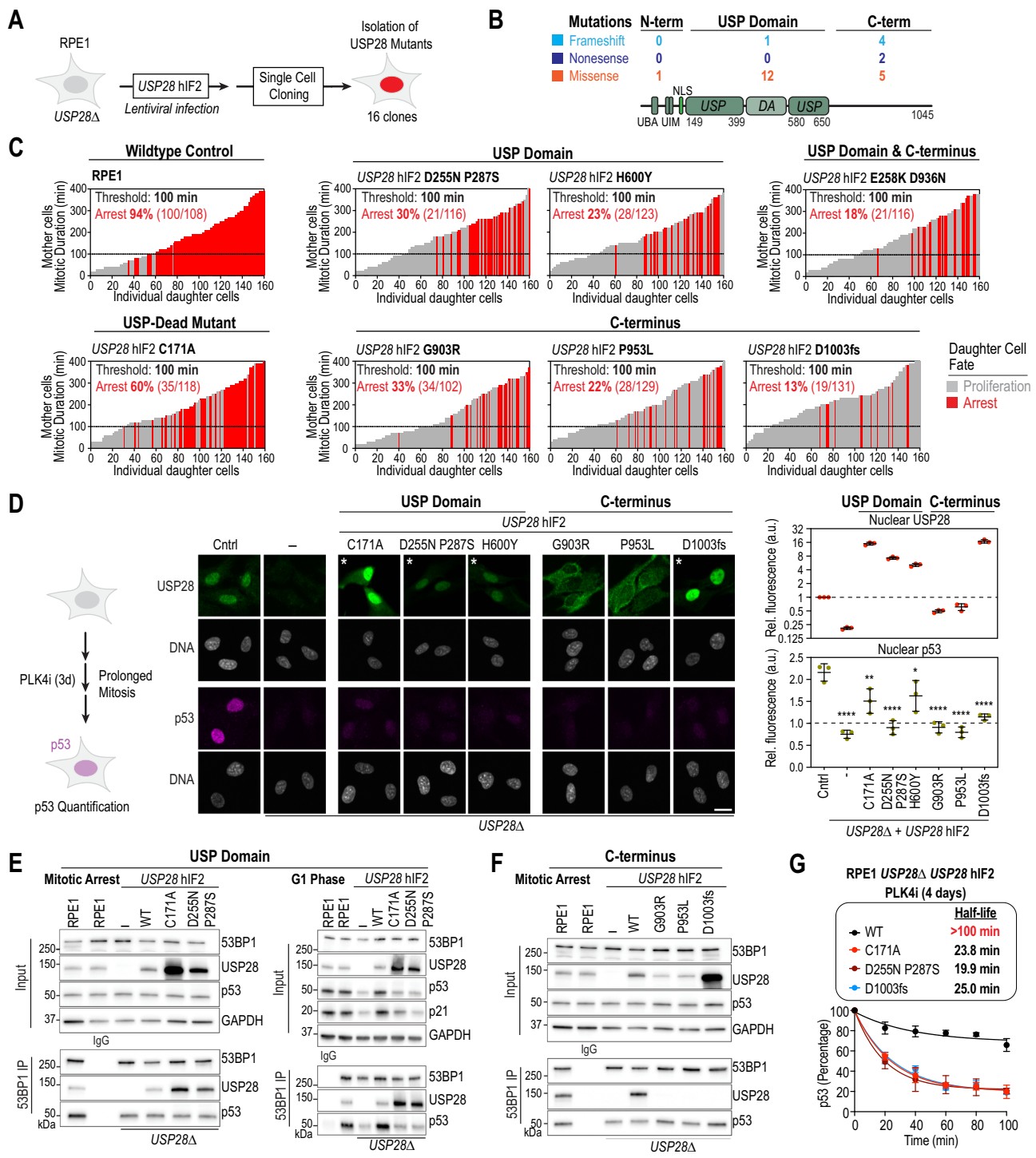

This suggests that the mutations impair USP28's deubiquitinase activity, which is essential for p53 stabilization[7]. Strikingly, the three mutants with distinct mutations in the C-terminal domain failed to interact with 53BP1, indicating that the C-terminus of USP28[hIF2] is required for its interaction with 53BP1 and subsequent p53 stabilization after prolonged mitosis (Fig. 3F).

We next assessed p53 stability in RPE1 *USP28Δ* cells expressing either wildtype or mutant USP28[hIF2] transgenes after PLK4-induced mitotic stress. Expression of the wildtype transgene markedly increased the p53 half-life from approximately 20 minutes in RPE1 *USP28Δ* cells to more than 100 minutes (Figs. 1F, 3G and S4A). In contrast, USP28[hIF2] variants carrying mutations in the USP domain or

C-terminus failed to stabilize p53, showing half-lives comparable to those in *USP28Δ* cells. Consistent with these results, cells expressing the mutant transgenes were unable to efficiently deubiquitinate p53, unlike cells expressing the wildtype transgene (Fig. S4B, C).

Taken together, these results demonstrate that both the USP domain and the C-terminal domain of USP28 are required for p53 stabilization following mitotic stress.

### The C-terminus and dimerization of USP28[hIF2] are required for interaction with 53BP1

One of the isolated mutants that failed to interact with 53BP1 harbored a frameshift mutation near the C-terminus of the coding sequence

**Fig. 3 | Identification of mutations in USP28 that desensitize cells to mitotic stress. A** Schematic showing isolation of cell clones expressing mutant *USP28*[hIF2] transgene. Mutations occurred spontaneously without applied stress. **B** Schematic of USP28 mutations, including missense, nonsense, and frameshift variants, grouped by location: N-terminal (aa 1–149), USP domain (aa 149–650), and C-terminal (aa 651–1045) (Fig. S3A). **C** Live imaging assay (as in Fig. 2A) in *USP28*-mutant clones. The RPE1 control graph is reused from Fig. 2A. n = 160 cells per graph, pooled from ≥3 independent replicates (RPE1: 80, 58, 22; C171A: 41, 23, 32, 23, 28, 13; D255N P287S: 45, 55, 18, 42; H600Y: 35, 62, 30, 33; E258K D936N: 6, 41, 67, 32, 14; G903R: 7, 70, 58, 25; P953L: 34, 32, 36, 22, 36; D1003fs: 88, 48, 24). The USP-dead control C171A is included. **D** Microscopy-based analysis of USP28 and p53 expression in RPE1, *USP28Δ*, and *USP28Δ* cells expressing *USP28*[hIF2] mutants. USP28 panels marked with an asterisk were displayed at 5× lower intensity (unadjusted images in Fig. S3C). Nuclei stained with Hoechst 33342. Representative cells are shown. Quantification of nuclear USP28 and p53 levels after 3 days of PLK4i

treatment. Mean ± SD. Normalized to untreated RPE1 cells. One-way ANOVA comparison to Cntrl (*USP28Δ*, ****P < 0.0001, CI (95%) = 0.9380 to 1.878; C171A, **P = 0.0051, CI (95%) = 0.1825 to 1.122; D255N P287S, ****P < 0.0001, CI (95%) = 0.7875 to 1.727; H600Y, *P = 0.0228, CI (95%) = 0.06401 to 1.004; G903R, ****P < 0.0001, CI (95%) = 0.7802 to 1.720; P953L, ****P < 0.0001, CI (95%) = 0.8937 to 1.833; D1003fs, ****P < 0.0001, CI (95%) = 0.5441 to 1.484). n = 3 independent replicates. Each point = mean of 500 cells. Scale bar: 10 μm. **E, F** 53BP1 immunoprecipitation and lysate analysis. **E** WT *USP28*[hIF2] and USP domain mutants (C171A, D255N/P287S). **F** WT *USP28*[hIF2] and C-terminal mutants (G903R, P953L, D1003fs). Inputs, soluble fractions. IP, immunoprecipitates. GAPDH, loading control. Representative from two independent experiments. **G** Cycloheximide chase assay measuring p53 half-life. Quantification from three independent experiments (see Fig. S4A). Protein decay fitted to exponential curve. Half-life = time to 50% loss. Mean ± SD. Source data are provided as a Source Data file.

---

(D1003fs). This mutation resulted in a 10-base pair deletion, leading to a C-terminally truncated protein (1007aa; full length is 1045aa) (Fig. S3D). To identify the minimal region of USP28[hIF2] required for interaction with 53BP1, we generated mutants expressing transgenes of shorter C-terminal truncations in RPE1 *USP28Δ* cells (Fig. 4A). We found that deletion of the last 13 amino acids was sufficient to disrupt the interaction with 53BP1, and this truncation failed to stabilize p53 following release from mitotic arrest (Fig. 4A, B). Taken together, our work identified four mutants that failed to interact with 53BP1 (Fig. 4C). The identified mutants have either a missense mutation (G903R, P953L) or a C-terminal truncation of 13 or more amino acids. The longer isoform USP28[hIF1] has an additional alpha helix between amino acids 769 and 800, which impairs the interaction with 53BP1. These results indicate that the predicted C-terminal domain (aa651–1045) is required for the interaction with 53BP1.

Next, we sought to determine the minimal region of USP28[hIF2] required for 53BP1 binding. As expected, the N-terminal region (aa1–650), including the USP domain and dimerization arm, was unable to mediate 53BP1 interaction (Figs. 4B and S5A). Surprisingly, the C-terminal region alone (aa651–1045) was also insufficient (Fig. 4B and S5A–D). Forcing nuclear localization of the C-terminal fragment with a nuclear localization signal (NLS, from nucleoplasmin) did not enhance 53BP1 binding (Figs. 4B, D and S5B–D). However, a larger fragment containing both the C-terminus and the ubiquitin-specific protease (USP) domain (aa157–1045) could interact with 53BP1 (Fig. 4B, D).

USP28 contains a unique USP domain interrupted by a dimerization arm (DA; aa400–579)[28,29]. An AlphaFold model of the USP domain reveals both structured and unstructured regions within this domain (Figs. 4E and S5E), closely resembling configurations of previously reported structures[28,29]. Interestingly, a construct containing only the dimerization arm but lacking the N-terminal portion of the USP domain, was sufficient to interact with 53BP1 (Fig. 4B, F). To determine whether dimerization is necessary for this interaction, we tested a USP28 dimerization mutant, V541E L545E (Fig. 4B, E, G)[28,29]. We found that this dimerization mutant was unable to interact with 53BP1. In contrast, an unstructured region (Δ460–520) within the dimerization arm was not necessary (Fig. 4B, E, G).

Our data suggest that 53BP1 either exclusively binds the C-terminus of dimerized USP28 or that the dimerization arms create an additional binding surface for 53BP1. To differentiate between these possibilities, we fused an inducible dimerization domain (DmrB) to the N-terminus of a C-terminal USP28 fragment (651–1045), which by itself was unable to interact with 53BP1. Strikingly, DmrB-induced dimerization of the C-terminal USP28 fragment drastically enhanced its interaction with 53BP1 (Fig. 4H–J and S5F). A similar effect was observed with a slightly longer C-terminal fragment (580–1045) (Fig. S5G–I). These results demonstrate that 53BP1 interacts exclusively with the dimerized form of the USP28[hIF2] C-terminus.

## PLK1 promotes the interaction between the USP28 C-terminus and 53BP1

The kinase PLK1 is essential for the interaction between USP28 and 53BP1 (Fig. 1I)[13]. Based on our findings, PLK1 could either regulate USP28 dimerization or directly promote the interaction between the USP28 C-terminus and 53BP1. To distinguish between these possibilities, we utilized the chemically enforced USP28 dimer, which functions independently of PLK1 (Fig. 4H–J). We found that PLK1 inhibition still impairs this interaction, suggesting that PLK1-dependent phosphorylation promotes the interaction between the USP28 C-terminus and 53BP1 rather than dimerization (Fig. S6A). Moreover, our results show that the interaction between 53BP1 and the chemically enforced dimer of the USP28 C-terminus remains stable after mitosis, mirroring the behavior of full-length USP28 (Fig. S6B).

To identify the relevant PLK1 target, we generated a phosphorylation-deficient mutant (16 A) based on predictions from GPS 6.0[33]. In this mutant, all high-confidence phosphorylation sites within the USP28 C-terminus were mutated to alanine (Fig. S6C). The 16 A mutant retained its ability to interact with 53BP1, indicating that USP28 itself is unlikely to be the direct PLK1 target. This suggests that either 53BP1 is the relevant PLK1 target, or that alternative mechanisms regulate the interaction between 53BP1 and USP28[34].

## UBA and UIM domains of USP28 are not required for p53 activation

The N-terminus of USP28 is not essential for its interaction with 53BP1, but it contains two distinct regions: the Ubiquitin-associated domain (UBA) and the Ubiquitin-interacting motif (UIM) (Fig. S7A). These domains are proposed to bind ubiquitinated proteins and may play a role in substrate recognition of USPs[30]. To evaluate whether UBA and UIM domains are necessary for p53 stabilization under mitotic stress, we isolated two single clones of truncation mutants lacking UBA (92–1045) or both UBA and UIM (157–1045) (Fig. S7). All clones, including those expressing full-length USP28 and the truncated variants, showed comparable expression levels. Unexpectedly, all tested mutants were able to stabilize p53 and induce p21 expression, similar to wildtype RPE1 cells. These findings suggest that UBA and UIM-mediated substrate recognition by USP28 is dispensable for p53 stabilization in response to mitotic stress.

## Cancer-associated mutations in USP28 impair mitotic stress response

USP28 is frequently mutated in cancer and is classified as tumor suppressor (Fig. 5A)[11,35]. Frameshift mutations in USP28 disrupt protein expression and render cancer cells insensitive to mitotic stress[13]. While frameshift mutations are known to impair USP28 function, most cancer-associated mutations in USP28 are missense mutations, substituting a single amino acid (Fig. 5A, B). We hypothesized that missense mutations in USP28 may desensitize cancer cells to mitotic

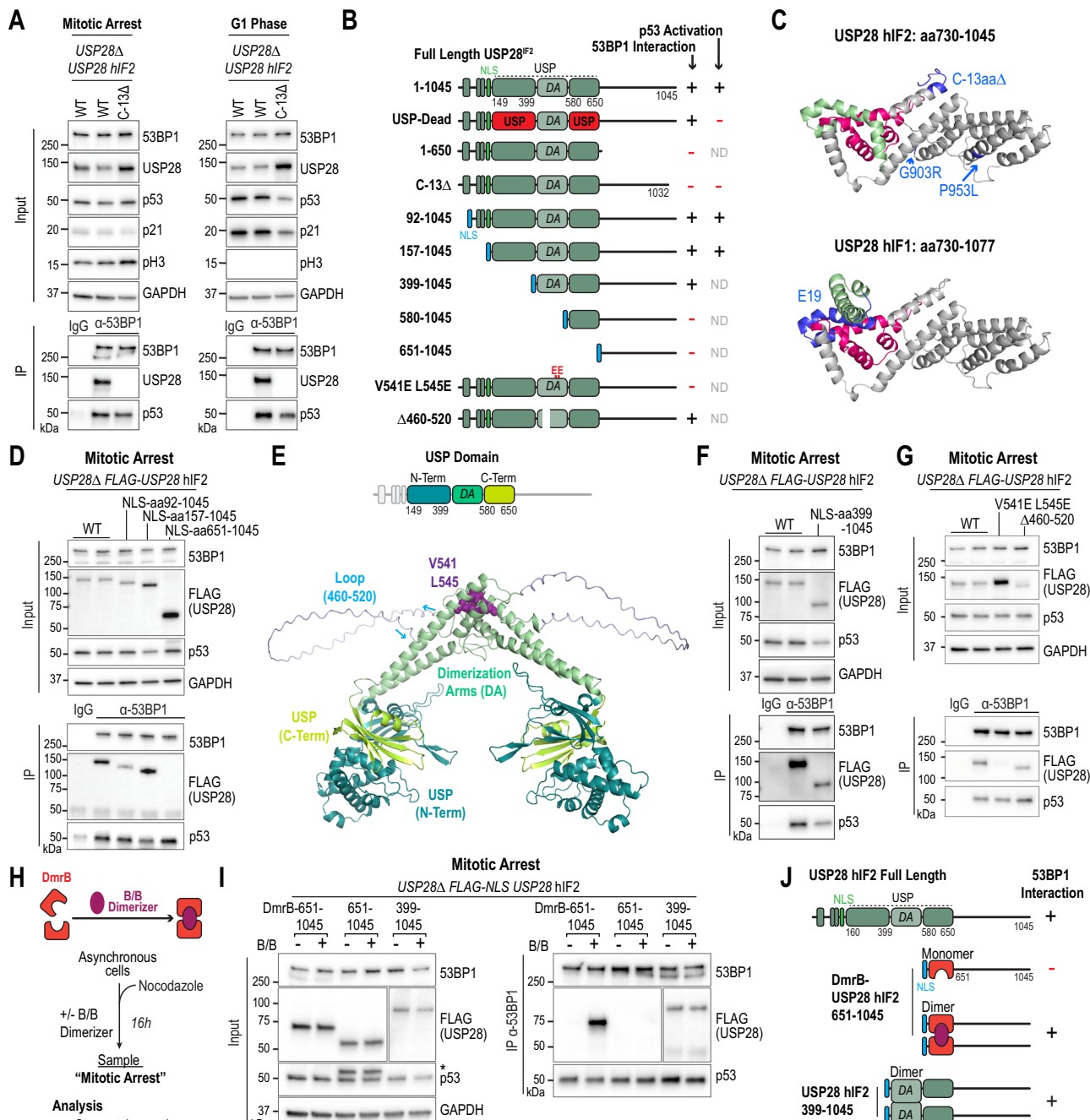

**Fig. 4 | The C-terminus and dimerization of USP28^hIF2 are required for interaction with 53BP1. A** 53BP1 immunoprecipitation and lysate analysis to assess complex formation with USP28 and p53 (Mitotic Arrest) and p53 stabilization (G1 Phase). *USP28Δ* RPE1 cells expressed C-terminally truncated *USP28*^hIF2 (Δ13). Inputs, soluble supernatants. IP, immunoprecipitates. GAPDH, loading control. Input samples derive from the same experiment but different gels for 53BP1, USP28, p53, GAPDH, and another for p21 were processed in parallel. Representative of three independent experiments. **B** Summary of USP28 mutants and their ability to bind 53BP1 and stabilize p53. NLS sequences are indicated (endogenous, green; nucleoplasmin-derived, blue). **C** AlphaFold model of the USP28 C-terminus showing isoform 2 mutations and the exon 19 alpha helix of isoform 1 interfering with 53BP1 binding. **D** 53BP1 immunoprecipitation in cells expressing WT or N-terminal truncated *USP28*^hIF2 variants (aa92–1045, aa157–1045, aa651–1045), all FLAG-tagged and NLS-fused. Inputs/IP/GAPDH as above. Samples derive from the same experiment but different gels for 53BP1, p53, GAPDH, and another for Flag-USP28 were processed in parallel. Representative of three independent experiments. **E** AlphaFold model of the USP28 dimer showing the USP domain,

dimerization arm (DA), critical residues (V541, L545) and unstructured loop (aa460–520). **F, G** 53BP1 immunoprecipitation in cells expressing *USP28*^hIF2 WT or mutants. **F** Construct lacking the N-terminal region including part of the USP domain. Samples derive from the same experiment but different gels for 53BP1, FLAG-USP28, GAPDH, and another for p53 were processed in parallel. **G** Constructs with DA mutations (V541E, L545E) or loop deletion (Δ460–520). Representative of two independent experiments. Inputs/IP/GAPDH as above. **H** Schematic of inducible dimerization domain. Asynchronous cells treated with Nocodazole (100 ng/ml, 16 h) and B/B dimerizer (100 nM). **I** Co-immunoprecipitation of dimerization-dependent 53BP1–USP28–p53 complexes. *USP28Δ* RPE1 cells expressed USP28 fragments (651–1045 or 399–1045) fused to DmrB-FLAG or control constructs. Low- and high-exposure blots were used for comparability (see Fig. S5F). Asterisk indicates residual FLAG signal on the p53 blot. Inputs/IP/GAPDH as above. Representative of two independent experiments. **J** Summary of (**H–I**) showing that USP28 dimerization is required for C-terminal 53BP1 binding. Source data are provided as a Source Data file.

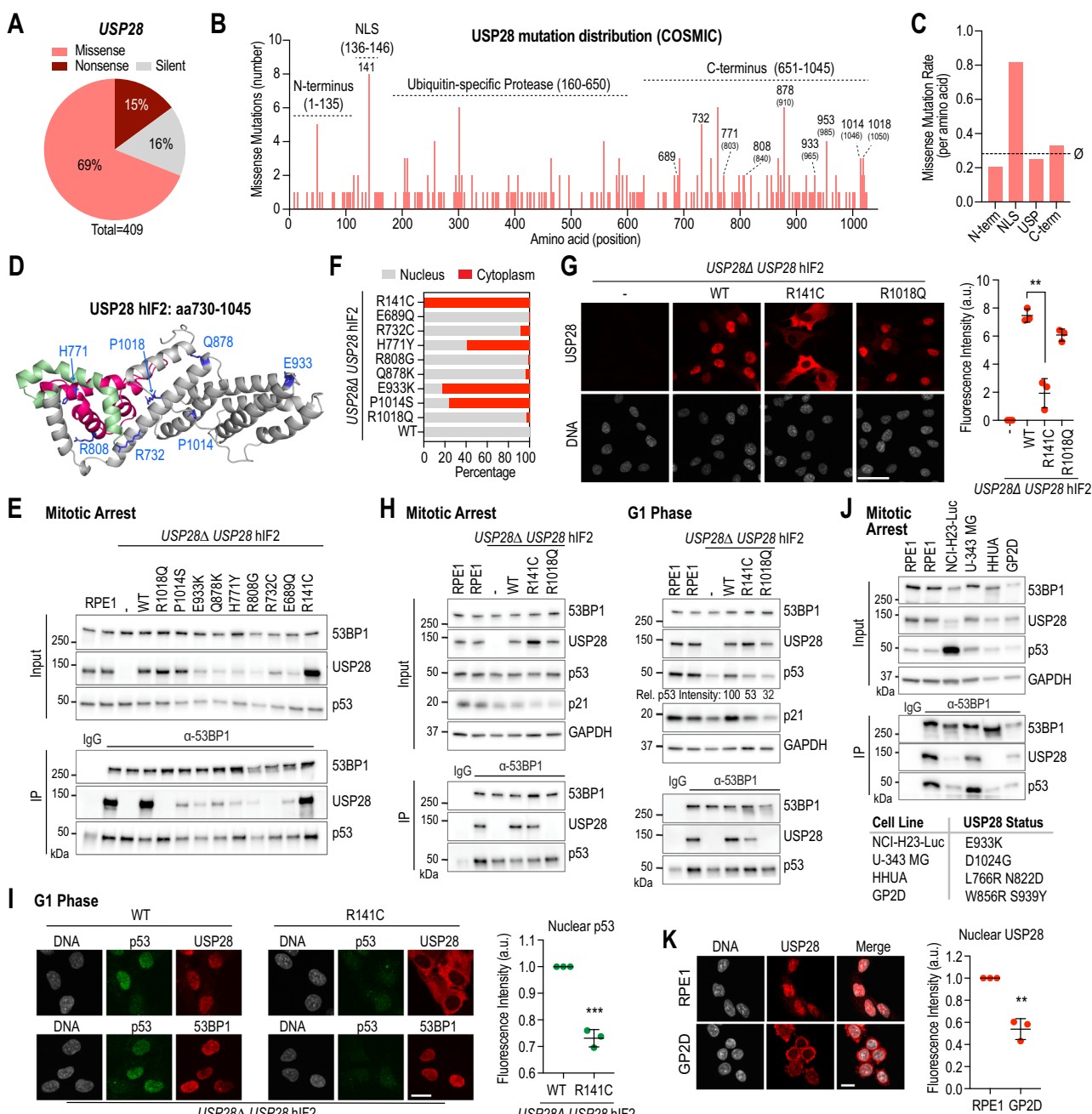

**Fig. 5 | Cancer-associated mutations in USP28 impair mitotic stress response.**
**A** Pie chart of USP28 point mutations in tumors (COSMIC)[35]. **B** Distribution of missense mutations across USP28hIF2 (USP28hIF1 positions are in brackets); regions quantified in (**C**) highlighted. **C** Missense mutation rate per amino acid in distinct domains; dotted line marks gene-wide average. **D** AlphaFold model of the USP28 C-terminus highlighting residues frequently mutated in cancer. **E** 53BP1 immuno-precipitation in *USP28Δ* RPE1 cells expressing WT or mutant *USP28*hIF2. Unmodified RPE1 served as control. Inputs, soluble supernatants. IP, immunoprecipitate. GAPDH, loading control. Representative of two independent experiments. **F** Quantification of nuclear vs. cytoplasmic localization for cancer-associated USP28 mutants (highlighted in B, D). Values represent the average of two inde-pendent replicates. n(WT) 99, 100; n(R1018Q) 109, 119; n(P1014S) 76, 199; n(E933K) 29, 36; n(Q878K) 67, 32; n(R808G) 48, 27; n(H771Y) 38, 33; n(R732C) 67, 48; n(E689Q) 49, 50; n(R141C) 69, 100. **G** Representative images showing localization of WT and mutant USP28. Graph shows total nuclear USP28 levels normalized to *USP28Δ*. Mean ± SD. Two-sided Student's *t*-test (**P = 0.0011, CI (95%) = −7.356 to

−3.699). n = 3 independent replicates. Each point = mean of 50 cells. Scale bar: 25 μm. **H** 53BP1 immunoprecipitation and lysate analysis in WT or mutant *USP28*hIF2 (R141C, R1018Q) to assess complex formation with USP28 and p53 (Mitotic Arrest) and p53 stabilization (G1 Phase). Inputs/IP/GAPDH as above. Representative of two independent experiments. **I** Representative immunostaining images of USP28 and 53BP1 after mitotic release. Graph shows total nuclear p53 levels normalized to WT. Mean ± SD. Two-sided Student's *t*-test (***P = 0.001, CI (95%) = −0.3210 to −0.2168). n = 3 independent replicates. Each point = mean of 60 cells. Scale bar: 10 μm. **J** 53BP1 immunoprecipitation in cancer-derived cell lines carrying C-terminal USP28 mutations. Inputs/IP/GAPDH as above. Representative of two independent experiments. **K** Representative USP28 immunostaining of RPE1 and GP2D cells. Graph shows total nuclear USP28 levels normalized to RPE1. Mean ± SD. Two-sided Student's *t*-test (**P = 0.0011, CI (95%) = −0.6128 to −0.3096. n = 3 independent replicates. Each point = mean of 50 cells. Scale bar: 10 μm. Source data are provided as a Source Data file.

stress, contributing to genome instability, which is a hallmark of cancer.

To test this hypothesis, we first analyzed the frequency distribution of missense mutations across the USP28 coding region. We found that amino acids in the C-terminus (aa651−1045) were more frequently mutated than those in the USP domain (aa160−650) or the N-terminus (aa1−135) (Fig. 5C). Notably, a single amino acid within the NLS (aa135−145), R141C, was the most mutated, suggesting that nuclear localization is critical for USP28 function.

To assess the impact of the most frequent C-terminal mutations and the NLS mutation, we expressed mutated USP28[hIF2] transgenes in RPE1 USP28Δ cells (Fig. 5B, D). We observed three distinct phenotypes. First, mutations between amino acids 689 and 933 led to protein destabilization, as evidenced by reduced expression levels (Fig. 5E), proposing that the reduced expression could desensitize to prolonged mitosis. Second, the R141C mutation within the NLS caused aberrant cytoplasmic localization (Fig. 5F, G). Intriguingly, several mutations in the C-terminus also resulted in exclusion from the nucleus, consistent with the cytoplasmic localization of previously described mutants (G903R, P953L) (Figs. 3D and 5F). Proline 953 is also often mutated in cancer (Fig. 5B), supporting the possibility that USP28 is excluded from the nucleus in some cancers. Third, the R732C and R1018Q mutants failed to interact with 53BP1 regardless of their nuclear localization (Fig. 5E−G), which further supports the possibility that cancer impairs mitotic stress response by interfering with mitotic stopwatch complex formation and function.

To investigate the functional consequences of these mutations, we examined p53 stabilization following prolonged mitosis. We arrested R141C (nuclear exclusion) and R1018Q (failure to interact with 53BP1) mutants in mitosis and subsequently released them into the next cell cycle (Fig. 5H). Both mutants failed to sufficiently stabilize p53 in G1 phase. The inability of the R1018Q mutant to activate p53 is likely due to the lack of interaction with 53BP1. While the R141C mutant formed a stable complex with 53BP1 and p53 during mitotic arrest (Fig. 5E, H), it failed to stabilize p53 after mitotic exit, which might be caused by the spatial separation of cytoplasmic USP28 and nuclear 53BP1 (Fig. 5I). This finding suggests that USP28 undergoes dynamic turnover between 53BP1-bound and unbound pools, thereby enabling spatial separation of USP28 and 53BP1 after mitotic exit.

To confirm that the tested mutations have similar defects in cancer, we identified four cancer-derived cell lines that have either one homogeneous or two heterogeneous missense mutation in the C-terminus of USP28 (Fig. 5J). We found that three cell lines (NCI-H23, HHUA, GP2D) had reduced level of USP28 expression, which is in line with our finding from transgenic RPE1 cells (Fig. 5E, J). Furthermore, we found that USP28 either failed to interact with 53BP1 (HHUA) or had a lower degree of interaction (NCI-H23) (Fig. 5J). Immunostaining further revealed that in GP2D cells, USP28 was unable to localize to the nucleus and instead accumulated prominently in the cytoplasm (Fig. 5K).

In conclusion, all tested cancer-associated missense mutations in the C-terminus of USP28 resulted in nuclear exclusion, failure to interact with 53BP1, or protein destabilization; phenotypes which we also observed in cancer-derived cell lines. These mutations prevent the activation of p53 following extended mitosis, highlighting their role in disrupting the mitotic stress response in cancer cells.

### Destabilization of USP28 reduces sensitivity to mitotic stress

To assess if the reduced USP28[hIF2] amount in cancer cells alters the sensitivity to prolonged mitosis, we generated single-cell clones with homogeneous but different expression levels of USP28[hIF2] wildtype transgene (Fig. 6A−C). We selected three clones: Clone 1 (#1), which exhibited slightly higher USP28[hIF2] expression than wildtype RPE1 cells; Clone 2 (#2), with expression levels similar to wildtype; and Clone 3 (#3), which expressed lower levels of USP28[hIF2] (Fig. 6B−D). Following a 16-hour mitotic arrest, all three clones demonstrated complex

formation between USP28[hIF2], 53BP1, and p53, though the abundance of USP28[hIF2] within these complexes correlated with the level of USP28[hIF2] expression (Fig. 6D).

To determine whether the 53BP1-USP28[hIF2] complexes could activate p53 in response to prolonged mitosis, we treated cells with PLK4i to induce centrosome depletion and thereby prolong mitosis (Fig. 6E)[9,36]. PLK4 inhibition does not affect the first mitosis. The second and following mitoses are moderately prolonged (50−120 minutes)[13]. Since RPE1 cells complete a cell cycle in approximately 20 hours, they typically experience one or two moderately prolonged mitoses within three days of treatment. After three days of PLK4i treatment, we analyzed p53 stability through immunostaining (Fig. 6F, G) and observed that p53 activation levels correlated with USP28[hIF2] expression levels (Fig. 6H).

To further investigate whether the sensitivity to mitotic stress depends on USP28 expression levels, we tracked individual cells by live cell imaging and assessed the response to single prolonged mitosis (Fig. 2A). Clone 1, with approximately 1.5-fold higher USP28[hIF2] expression than wildtype cells, showed increased sensitivity to mitotic arrest, while Clone 3, with roughly half the wildtype USP28[hIF2] level, exhibited reduced sensitivity (Fig. 6I). Specifically, the threshold for inducing cell cycle arrest in response to prolonged mitosis was approximately 100 minutes in wildtype cells, 70 minutes in Clone 1, 90 minutes in Clone 2, and 150 minutes in Clone 3. These findings indicate that cellular sensitivity to mitotic stress is modulated by the expression level and stability of USP28[hIF2], establishing USP28 as a limiting factor for mitotic stress response.

## Discussion

USP28 has been described as a tumor suppressor and oncogene[11,37]. Here, we demonstrate that in both untransformed and cancerous p53-wildtype cells, USP28 primarily functions as a tumor suppressor by stabilizing p53 and downregulating MYC in response to mitotic stress (Fig. 1A−D). Unexpectedly, we found that in some cancer cell lines USP28 increases baseline p53 levels (Fig. S1A). In these cell lines, USP28 deletion not only dampens the response to mitotic stress but also weakens the response to DNA damage (Fig. 1D). We demonstrate that p53 stabilization occurs through USP28-dependent deubiquitination, thereby resolving a longstanding debate regarding the underlying mechanism (Fig. 1E−G)[7,8,38]. Furthermore, we identified molecular mechanisms of USP28 that are required for p53 stabilization following mitotic stress (Figs. 2−4) and demonstrate that these mechanisms are frequently disrupted in cancer (Figs. 5−7). These findings highlight the tumor-suppressive roles of USP28 and may have important implications for therapeutic strategies.

Using transgene expression, we found that p53 stabilization is mediated by the shorter isoform 2 of USP28, whereas the longer isoform 1−often considered the canonical isoform−does not have this function (Fig. 2C−I). The two isoforms differ by 32 amino acids that are expressed from exon 19. AlphaFold modeling predicted that this region forms an alpha helix (Fig. 2E). A possible explanation is that the additional 32 amino acids of the longer isoform 1 disrupt the interaction surface between the C-terminus of USP28 and 53BP1. In line with this model, we identified several point mutations (G903R and P953L) and truncation mutations in the same C-terminal region of USP28 (D1003fs, C-Δ13) that impair the interaction with 53BP1 (Fig. 3F; 4A) and consequently fail to stabilize p53 in response to mitotic stress (Figs. 3D, G and 4A). Surprisingly, isoform 2 of USP28 is more dominantly expressed than isoform 1 across 26 cancer cell lines from 10 different tissues (Fig. 2G, H and S2F, H, I). Additionally, we observed a similar expression pattern in various mouse organs, including liver, kidney, heart, muscle, and lung (Fig. 2I and S2G). The brain is the only organ identified in mice in which the longer isoform or other Exon 19 expressing isoforms were predominantly expressed. These findings suggest that the shorter isoform (USP28[hIF2]; USP28[mIF1]) is the canonical

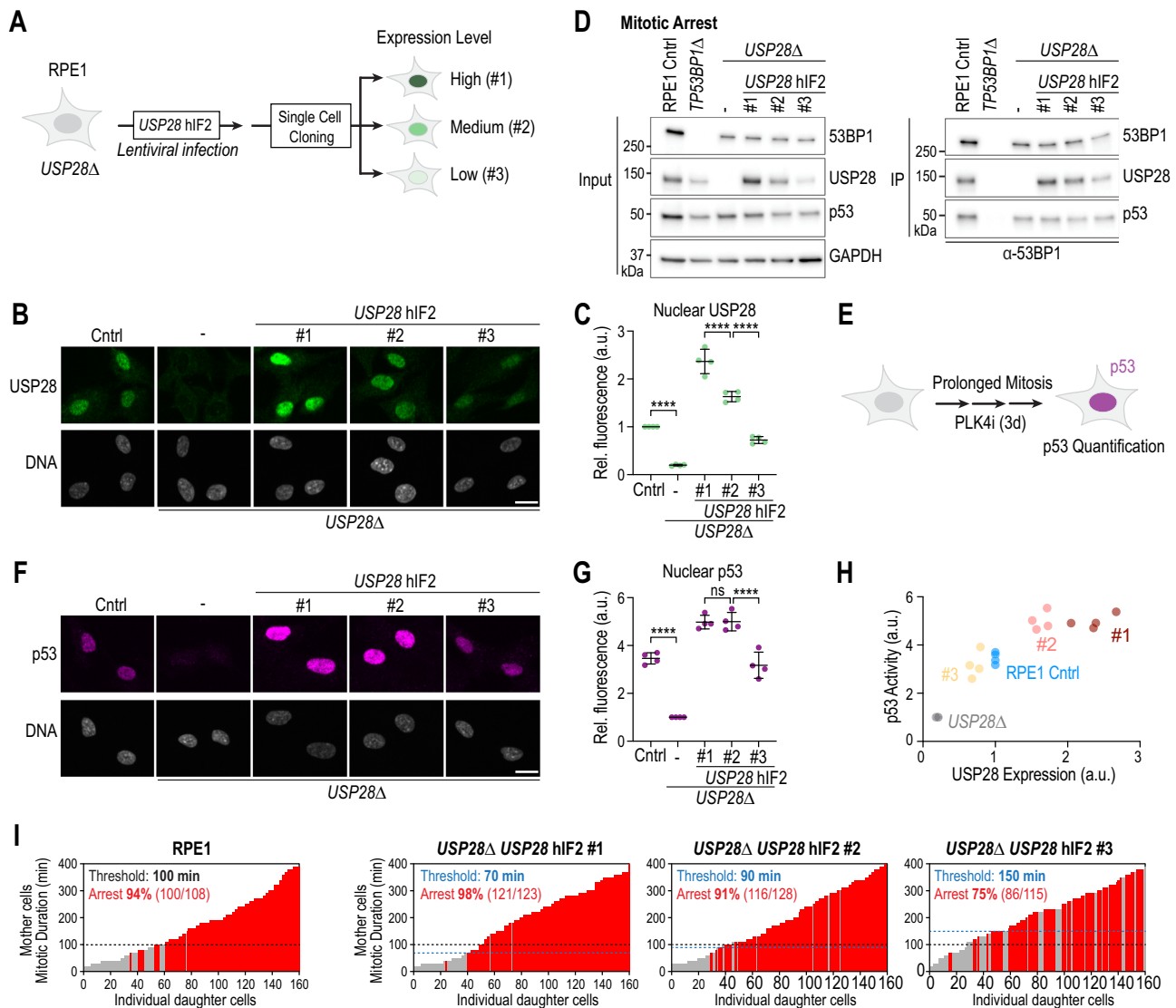

**Fig. 6 | USP28 confers concentration-dependent sensitivity to prolonged mitosis. A** Schematic of single-cell clone isolation expressing the *USP28*$^{hIF2}$ WT transgene. Clones #1, #2, #3 show homogeneous expression. **B** Representative RPE1 cells expressing endogenous USP28 (Cntrl) or varying *USP28*$^{hIF2}$ levels. Scale bar: 10 µm. **C** Quantification of USP28 staining from (**B**). Mean ± SD. One-way ANOVA: Cntrl vs. *USP28Δ*, ****P < 0.0001, CI (95%) = 0.5225 to 1.082; USP28 #1 vs. USP28 #2, ****P < 0.0001, CI (95%) = 0.4557 to 1.015; USP28 #2 vs. USP28 #3, ****P < 0.0001, CI (95%) = 0.6275 to 1.187. n = 4 independent replicates. Each point = mean of 2500 cells. **D** 53BP1 immunoprecipitation in clones expressing different *USP28*$^{hIF2}$ levels to assess complex formation. Inputs are soluble supernatants. IP, immunoprecipitate. GAPDH, loading control. Representative of two independent experiments. **E** Schematic of assay assessing sensitivity to prolonged mitosis. Cells were treated with PLK4i for 3 days and stained for p53 activation. **F** Representative images showing p53 expression following prolonged mitosis. Scale bar: 10 µm. **G** Quantification of p53 staining from (**F**). Mean ± SD. One-way ANOVA: Cntrl vs. USP28Δ, ****P < 0.0001, CI (95%) = 1.714 to 3.211; USP28 #1 vs. USP28 #2, P = > 0.9999, CI (95%) = −0.7703 to 0.7264; USP28 #2 vs. USP28 #3, ****P < 0.0001, CI (95%) = 1.075 to 2.572. n (Cntrl) 4; n (*USP28Δ*) 4; n (*USP28Δ USP28*$^{hIF2}$#1) 4; n (*USP28Δ USP28*$^{hIF2}$#2) 4; n (*USP28Δ USP28*$^{hIF2}$#3) 4. Each point = mean of 2500 cells. **H** Correlation of USP28 levels (from C) with p53 stabilization (from G) after prolonged mitosis. Each point = mean of independent replicates. **I** Imaging-based assay (as in Fig. 2A) examining how USP28 expression affects the mitotic duration threshold for cell cycle arrest. RPE1 control graph reused from Fig. 2A. n = 160 cells per graph, pooled from ≥3 independent replicates (RPE1: 80, 58, 22; *USP28* hIF2 #1: 33, 35, 38, 24, 30; *USP28* hIF2 #2: 61, 50, 59; *USP28* hIF2 #3: 64, 66, 30). Source data are provided as a Source Data file.

isoform, while the longer isoform (*USP28*$^{hIF1}$; *USP28*$^{mIF2}$) is context-specific. Further studies are needed to explore the precise function of the longer isoform (*USP28*$^{hIF1}$; *USP28*$^{mIF2}$) and its expression in different cell types.

We further investigated the minimal region required for the interaction between USP28 and 53BP1. Surprisingly, while the C-terminus of USP28 is required for binding to 53BP1, it is not sufficient on its own (Figs. 4B, D and S5A, B). The minimal USP28 fragment that was able to interact with 53BP1 contained the USP domain but lacked the N-terminal domain, which includes ubiquitin-binding motifs UBA

and UIM and an NLS (Fig. 4D). The USP domain of USP28 is interrupted by a dimerization arm-specific to USP25 and USP28 (Fig. 4E)[28,29]. Disruption of dimerization through two mutations (V541E, L545E) impaired USP28's interaction with 53BP1, suggesting that USP28 must form a dimer to interact with 53BP1 (Fig. 4B, G). These results suggest two possible mechanisms: either dimerization of USP28 is required for the interaction between the C-terminus of USP28 and 53BP1, or the dimerization arms create a second binding surface for 53BP1. Using inducible dimerization domains (DmrB), we were able to replace the function of the dimerization arm, showing that dimerization of the

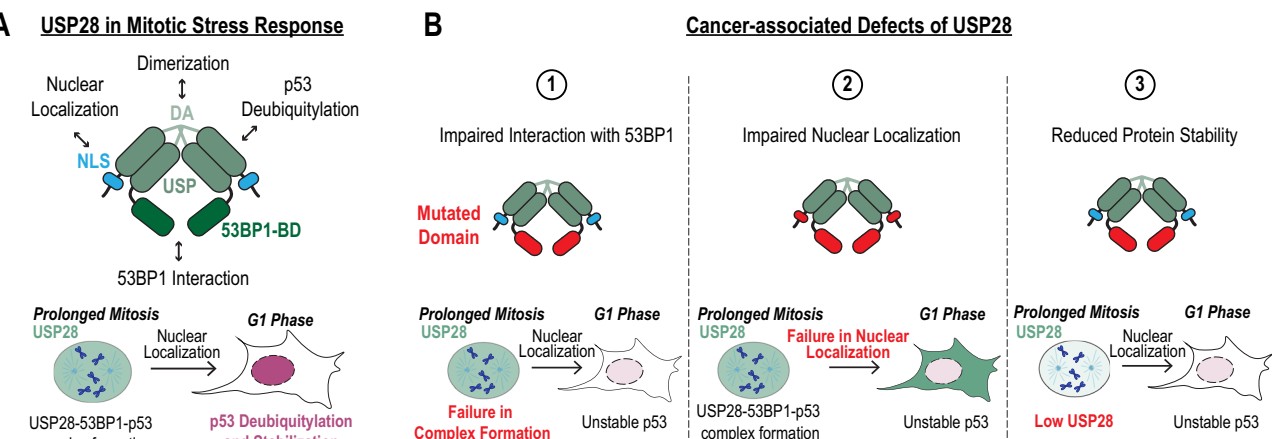

**Fig. 7 | Tumor-suppressive mechanisms of USP28. A** Model illustrating USP28 dimer structure and essential functions required for prolonged mitosis-mediated p53 stabilization. DA dimerization arm, USP ubiquitin-specific protease, 53BP1-BD 53BP1-binding domain, NLS nuclear localization sequence. **B** Schematic summarizing cancer-associated functional impairments caused by USP28 mutations: 1. Impaired interaction with 53BP1 (mutations in the 53BP1-BD). 2. Impaired nuclear localization (mutations in the NLS or 53BP1-BD). 3. Reduced USP28 stability (mutations in the 53BP1-BD).

C-terminus of USP28 is sufficient for 53BP1 binding (Figs. 4H–J and S5F–I). Furthermore, we found that the interaction of the USP28 C-terminus and 53BP1 is PLK1-dependent and critical for the transfer of mitotic stress signaling to the daughter cells (Fig. S6). Our findings raise new questions about the stoichiometry of the mitotic stopwatch complex. Previous studies have shown that 53BP1 forms oligomers[39,40] and p53 exists as a tetramer[41], which implies that the mitotic stopwatch complex may have a higher structural complexity.

USP28 is frequently impaired in cancer (Fig. 5A). Cancer-associated missense mutations in USP28 prevent p53 stabilization following mitotic stress. We identified five key features of USP28 essential for the mitotic stress response: the USP catalytic domain, 53BP1 interaction, dimerization, protein stability, and nuclear localization (Fig. 7A). Notably, defects in three of these features correspond to frequently occurring missense mutations in cancer (Fig. 7B). We found that the two out of ten (20%) tested cancer-associated mutations (R732C and R1018Q) impair the interaction with 53BP1 and fail to stabilize p53 following mitotic stress, highlighting the importance of this interaction (Fig. 5E, H). Several cancer-associated mutations in the C-terminus also result in reduced USP28 stability, which likely dampens the mitotic stress response in cancer cells (Figs. 5E and 6). The most frequent mutation in USP28 occurs within the NLS domain (Fig. 5B), impairing nuclear localization and the mitotic stress response (Fig. 5F–I). While nuclear exclusion does not prevent mitotic stopwatch complex formation, it impairs p53 stabilization in the G1 phase, likely because USP28 remains in the cytoplasm while 53BP1 is confined to the nucleus (Fig. 5H, I). This suggests that nuclear exclusion of USP28 can lead to dissociation of the mitotic stopwatch complex. Interestingly, mutations in the C-terminus also cause nuclear exclusion, but the underlying mechanism remains unclear, as no NLS sequence is detected in this region.

Several studies have identified USP28 as an oncogene in specific cancer contexts[1,2,4,42,43]. A recent study provided mechanistic insight into this function, showing that DNA damage activates ATM, which triggers the disassembly of USP28 dimers into monomers[3]. These monomeric forms then stabilize MYC, promoting oncogenic proliferation [3]. In contrast, our findings reveal a distinct pathway in which mitotic stress enhances the interaction between USP28 and 53BP1, leading to the stabilization of p53 and the downregulation of MYC. We further observed that cancer cells frequently disrupt the USP28−53BP1 interaction, potentially as a strategy to bypass p53-mediated cell cycle arrest or apoptosis (Fig. 5). Notably, USP28 mutants found in cancer may retain the ability to stabilize MYC, thus maintaining oncogenic signaling. Consistent with this, mutations are more frequently found in the C-terminal region of USP28−which mediates 53BP1 binding−than in the catalytic USP domain required for MYC stabilization (Fig. 5C). Given that USP28-mediated activation of p53 is critical for mounting an effective response to mitotic stress response and in some cancers to DNA damage response, disruption of its interaction with 53BP1 could contribute to therapeutic resistance, particularly against anti-mitotic and DNA-damaging agents.

A wide spectrum of cellular defects and environmental insults−including mitotic machinery dysfunction, DNA replication stress, aneuploidy, heat and osmotic stress, oncogene activation, irradiation, increased cell size, and viral infections−can lead to mitotic errors or DNA damage, ultimately promoting genomic instability[14,44−56]. We propose that USP28 acts as a central node in the cellular stress response network, particularly in the context of mitotic and DNA damage stress. By fine-tuning the cellular threshold for responding to both intrinsic and extrinsic stressors, USP28 may play a key role in maintaining genomic stability and tissue integrity.

## Methods

### Chemical inhibitors
Chemical inhibitors and their working concentrations were as follows: B/B Homodimerizer (100 nM; Takara), BI2536 (PLK1 inhibitor; 100 nM; MedChemExpress), Centrinone (PLK4 inhibitor, LCR-263; 150 nM; MedChemExpress), Cycloheximide (100 μg/ml; Sigma-Aldrich), Doxorubicin (1–1000 nM; Cell Signaling), Monastrol (100 μM; Tocris Bioscience), and Nocodazole (0.1 μg/ml; Sigma-Aldrich).

### Antibodies
Antibodies used in this study were obtained from commercial sources, with working concentrations indicated: 53BP1 (1:5000, Novus Biologicals, Cat# NB100-304, RRID:AB_10003037); Flag M2 (1:1000, Sigma-Aldrich, Cat# F1804-200UG, RRID:AB_262044); GAPDH (1:1000, Cell Signaling Technology, Cat# 5174, RRID:AB_10622025); Histone H3.3 (1:1000, Abcam, Cat# ab5176, RRID:AB_304763); Myc 9E10 (1:1000, Sigma-Aldrich, Cat# M4439, RRID:AB_439694); p21 (1:1000, Cell Signaling Technology, Cat# 2947, RRID:AB_823586); p53 (1:1000, Santa Cruz Biotechnology, Cat# sc-126, RRID:AB_628082); phospho-Histone H2A.X (1:2000, Millipore, Cat# 05-636, RRID:AB_309864); Ubiquitin (1:1000, Cell Signaling Technology, Cat# 3936, RRID:AB_331292); USP28 (1:1000, Abcam, Cat# ab126604, RRID:AB_11127442); USP28

(1:100, Sigma-Aldrich, Cat# HPA006778, RRID:AB_1080520); and Rabbit IgG (1:5000, Vector Laboratories, Cat# I-1000-5). Secondary antibodies were obtained from GE Healthcare and Jackson ImmunoResearch.

## Cell lines

A full list of cell lines used in this study is provided in Supplementary Data 1. The following were obtained from the American Type Culture Collection (ATCC): RPE1 (hTERT RPE-1), MCF7, CHP212, G401, A375, HCT116, RKO, U2OS, SJSA1, 769 P, H460, SH-SY5Y, N2A, and MP41. CHP134 and Mel-202 cells were sourced from Sigma-Aldrich (ECACC general collection), while G402, A549, LU99, Caki-1, ONDA9, A172, and OVTOKO cell lines were acquired from the Japanese Collection of Research Biosources (JCRB) Cell Bank. LOX-IMVI cells were obtained from the NCI-60 collection. All cell lines were maintained in their recommended media at 37 °C in a humidified atmosphere containing 5% $CO_2$, supplemented with 100 IU/ml penicillin and 100 μg/ml strepto-mycin. To induce mitotic arrest, cells were exposed to either 100 ng/ml nocodazole (Sigma-Aldrich) or 100 μM monastrol (Tocris Bioscience) for the indicated time periods.

The generation of the RPE1 USP28Δ line has been described previously[13]. o engineer USP28Δ RPE1 derivatives, lentiviral constructs encoding the indicated transgenes (see Supplementary Data 2) were introduced, including H2B-mRFP under the EF1α promoter and USP28 mutant variants under the UbC promoter. Lentiviral particles were produced by transfecting HEK-293T cells with the plasmids using the Lenti-X Packaging Single Shots system (Clontech, Cat# 631276). Culture supernatants containing virus were collected 48 h post-transfection and applied to target cells in the presence of 8 μg/ml polybrene (EMD Millipore). Stable populations were established through antibiotic selection (Neomycin, 400 μg/ml) or fluorescence-activated cell sorting (FACS). Single-cell clones expressing USP28 transgenes were isolated in 96-well plates and screened by immuno-fluorescence. Genomic DNA was extracted using the Quick-DNA Microprep Kit (ZYMO Research, Cat# D3021), and USP28 transgene sequences were verified by Sanger sequencing to identify specific mutations.

## Plasmid construction

All plasmids used in this study are described in Supplementary Data 2. USP28 mRNA Isoform-2 (NCBI reference sequence: NM_001346258.2), which lacks Exon 19 compared to the canonical isoform-1 mRNA sequence (NCBI reference sequence: NM_020886.4) cloned into len-tiviral expression vectors. We optimized the UbC promoter length (deletion of 245 bp from the 5′end of the promoter region) to reduce USP28 expression. This plasmid served as the backbone for generating USP28 mutant transgene constructs using site-directed mutagenesis with Phusion® High-Fidelity DNA Polymerase (NEB, Cat#M0530L). Cluster mutants were synthesized (gBlocks, IDT) and cloned into the vector.

## Immunofluorescence

For imaging experiments, ~5000 cells were seeded per well in 96-well imaging plates (SCREENSTAR, Cat# 655866) and cultured for 24 h prior to fixation. Cells were fixed with 100 μl of ice-cold methanol at −20 °C for 7 min, followed by two washes with PBS containing 0.1% Triton X-100. Blocking was carried out in PBS supplemented with 2% BSA, 0.1% Triton X-100, and 0.1% sodium azide for 2 h at 37 °C or overnight at 4 °C. After blocking, cells were incubated for 1–2 h with the indicated primary antibodies prepared in fresh blocking buffer (concentrations listed above). Following three washes with PBS/Triton buffer, cells were incubated for 1 h with the appropriate secondary antibodies and counterstained with Hoechst 33342 to visualize DNA. Finally, cells were washed three additional times before imaging. Image acquisition was performed using a CellVoyager CQ1 spinning

disk confocal microscope (Yokogawa Electric Corporation) equipped with a ×40 objective (0.95 NA) and a 2000 × 2000 pixel sCMOS camera (ORCA-Flash4.0V3, Hamamatsu Photonics). CQ1 software was used for image collection.

## USP28 transgene activity

To assess the function of wild-type and mutant USP28 variants, USP28Δ RPE1 cells stably expressing the respective transgenes were exposed to either the PLK4 inhibitor Centrinone (150 nM) or DMSO control for 3–4 days[36]. On day 2 or 3 of treatment, ~7000 cells were seeded into 96-well imaging plates and maintained under the same treatment conditions. Cells were fixed on day 3 or 4, followed by staining with Hoechst to visualize DNA and immunolabeling for USP28 and p53. Images were collected using a CQ1 spinning disk confocal microscope (Yokogawa Electric Corporation) as described previously, with acquisition performed through CQ1 software. Quantification of signal intensities was carried out using the Yokogawa Pathfinder ana-lysis platform.

## Live cell imaging

Live cell imaging was performed on the CellVoyager CQ1 spinning disk confocal system (Yokogawa Electric Corporation) equipped with a 40 × 0.95 NA U-PlanApo objective and a 2000 × 2000 pixel sCMOS camera (ORCA-Flash4.0V3, Hamamatsu Photonics) at 37 °C and 5% $CO_2$. Image acquisition and data analysis were performed using CQ1 software and ImageJ, respectively.

## Mitotic stopwatch assay

All cell lines used for imaging experiments were engineered to stably express H2B-RFP (see Supplementary Data 1). Cells were seeded into 96-well SCREENSTAR plates (Greiner, Cat# 655866) at a density of 2000–4000 cells per well one day prior to imaging. On the day of the experiment, asynchronous cultures were imaged in the presence of 100 μM Monastrol for 4–6 h. H2B-RFP signals were collected as 5 × 2 μm z-stacks in the RFP channel (25% laser power, 150 ms exposure) at 10-min intervals. Under Monastrol treatment, cells entered mitosis at variable times and were arrested in prometaphase, resulting in a population of mother cells that experienced different mitotic dura-tions. Following drug washout, the completion of mitosis was mon-itored every 10 min for 2 h, and the behavior of the resulting daughter cells was tracked at 20-min intervals for 48–72 h. Daughter cell fates were categorized as "arrest," "death", or "proliferation". The mitotic stopwatch threshold was defined as the minimum mitotic duration at which >50% of daughter cells entered arrest.

## Competition assays

Wildtype cells and a pool of heterogeneous USP28 or TP53BP1 knock-out cells were mixed and seeded into 10 cm plates at 100,000–300,000 cells/plate and treated with PLK4i (150 nM), Dox-orubicin (10 nM) or DMSO as a control. Cells were maintained for 8 days and passaged to avoid complete confluence. After 8 days cells were harvested, and genomic DNA purified. The genomic region tar-geted in USP28 or TP53BP1 were sequenced, and the knockout ratio was calculated by the indel decomposition software TIDE. The ratios PLK4i/DMSO and Doxorubicin/DMSO were calculated and blotted for each cell line.

## Proliferation assays

For proliferation measurements, 25,000 cells were seeded per well into 6-well plates in triplicate and exposed to the indicated inhibitors or DMSO as control. At 96-hour intervals, cells were collected, coun-ted, and, in passaging experiments, re-plated at the same initial density (25,000 cells/well). Cell numbers were determined using a TC20 automated cell counter (Bio-Rad).

## MYC and p53 stability assay

For the MYC stability analysis, 100,000 cells were plated into a 6-well plate. After 24 hours, cycloheximide was added to the cell culture medium at a concentration of 100 µg/ml. Cells were then harvested at the indicated time points. Following this, the cells were lysed, and 5 µg of the lysate was immunoblotted using the specified antibodies. To determine the half-life of p53 in RPE1 50,000 cells treated with either DMSO or PLK4 inhibitor (PLK4i) for a duration of four days prior to cycloheximide treatment.

## In vivo ubiquitination assay

For the in vivo ubiquitination assay[57], RPE1 cells grown in 15 cm dishes were subjected to treatment with the PLK4 inhibitor Centrinone (150 nM) for 4 days. Following the treatment, cells were harvested, and total protein was extracted using 200 µl of denaturing lysis buffer containing 50 mM Tris-HCl (pH 7.4), 0.5% SDS, and 70 mM β-mercaptoethanol. Lysis was achieved through vortexing, sonicating, and boiling the samples for 15 minutes at 95 °C. Subsequently, the lysates were combined with 800 µl of CHAPS buffer composed of 0.5% CHAPS, 10 mM Tris-HCl (pH 7.5), 1 mM MgCl2, 1 mM EGTA, 5 mM β-mercaptoethanol, 10% glycerol, 10 µM MG132, and 50 µM PR-619 to facilitate protein solubility. The lysates were incubated with an anti-p53 antibody (Santa Cruz Biotechnology, Cat# sc-126, RRID: AB_628082) for 2 hours at 4 °C. Following this, Protein A magnetic beads (Thermo Fisher Scientific, Cat# 88845) were added, and the mixture was incubated for an additional hour at the same temperature. The beads were then subjected to five washes with CHAPS lysis buffer to remove non-specific binding. Subsequently, the beads were boiled in 50 µl of 2 × SDS sample buffer to elute the proteins. The resulting samples were analyzed through SDS-PAGE, and ubiquitinated p53 was detected via immunoblotting with an anti-Ubiquitin antibody. Samples were quantified using ImageJ software. TP53-sh served as background control. Samples were corrected by background signal (TP53-sh) and normalized to the WT control.

## Artificial homodimerization assay of USP28

The inducible homodimerization domain DmrB (Takara) with an N-terminal FLAG tag was amplified by PCR and cloned into USP28-expressing lentiviral constructs upstream of USP28 fragments (*UbC^pro*-*3xFLAG-DmrB-USP28-(aa580–1045); UbC^pro-3xFLAG-DmrB-USP28-(aa651–1045)*). The transgenes were stably integrated into the *USP28Δ* RPE1 cell line genome using lentiviral constructs. To induce USP28 homodimerization, cells were treated with 100 nM of B/B Homo-dimerizer along with Nocodazole (100 ng/ml) for 16 hours. After collecting the treated mitotic cells, they were subjected to immunoprecipitation.

## Immunoblotting

Cells were grown in 15 cm dishes and harvested at ~80% confluency. Cell pellets were lysed by sonication in buffer containing 20 mM Tris-HCl (pH 7.5), 50 mM NaCl, 0.5% Triton X-100, 5 mM EGTA, 1 mM dithiothreitol, and 2 mM MgCl2, supplemented with protease and phosphatase inhibitors (Thermo Fisher Scientific). Lysates were cleared by centrifugation at 15,000 × g for 15 min at 4 °C, and supernatants were collected and stored at −80 °C. Protein concentrations were determined using the Bio-Rad Protein Assay, and 5–10 µg of protein per sample was resolved on Mini-PROTEAN gels (Bio-Rad) and transferred to PVDF membranes using the Trans-Blot Turbo transfer system (Bio-Rad). Membranes were blocked and probed with antibodies in TBS-T containing 5% non-fat milk. Detection was carried out with HRP-conjugated secondary antibodies (GE Healthcare) and SuperSignal West Femto substrates (Thermo Fisher Scientific). Chemiluminescent signals were captured on a ChemiDoc MP imaging system (Bio-Rad).

## Immunoprecipitation

For mitotic immunoprecipitation assays, cells were arrested in mitosis by treatment with nocodazole (100 ng/ml; 0.33–0.66 µM) for 8 or 16 h. A total of $1–2 × 10^6$ cells were collected, washed once in PBS, and resuspended in lysis buffer containing 20 mM Tris-HCl (pH 7.5), 50 mM NaCl, 0.5% Triton X-100, 5 mM EGTA, 1 mM dithiothreitol, 2 mM MgCl2, and an EDTA-free protease inhibitor cocktail (Roche). Cells were lysed in an ice-cold sonicating water bath for 5 min, and insoluble material was removed by centrifugation at 15,000 × g for 15 min at 4 °C. Protein concentrations were quantified, and 1–2 mg of lysate was incubated with anti-53BP1 antibody (Novus Biologicals, Cat# NB100-304, RRI-D:AB_10003037) for 2 h at 4 °C, followed by incubation with Protein A magnetic beads (Thermo Fisher Scientific, Cat# 88845) for 1 h at 4 °C. Beads were washed five times with lysis buffer and resuspended in SDS sample buffer. Immunoblotting was carried out by loading equal volumes on Mini-PROTEAN gels (Bio-Rad), transferring to PVDF membranes with the Trans-Blot Turbo system (Bio-Rad), and probing with antibodies in TBS-T containing 5% nonfat dry milk, as described above.

For immunoprecipitation from G1-phase cells following mitotic arrest, cells were first treated with nocodazole (100 ng/ml) for 8 or 16 h. Mitotic populations were then isolated by washing four times with PBS and replating onto 15 cm dishes. Cells were allowed to progress into G1 for 6 h before harvesting, after which immunoprecipitation was performed as outlined above.

## RT-PCR analysis

Total RNA was prepared by lysing $1 × 10^7$ cells and 100 mg of mouse tissue with RNeasy Plus Kit (Qiagen) and TRIzol (Thermo Fisher Scientific) respectively. A total of 500 ng of RNA was used to generate cDNA by using a PrimeScript™ II 1st strand cDNA Synthesis Kit (TAKARA BIO, Japan, 6210 A) with oligo dT primer. Subsequently, 1 µl of 1st strand cDNA was used as a template for PCR analysis. PCR products were analyzed by electrophoresis to confirm amplification. The primer sequences are listed in Supplementary Data 3.

## Figure legend compliance statement

All figure legends comply with *Nature Communications* reporting standards and include sufficient experimental detail to allow interpretation without reference to the main text. For all immunoblots, input, immunoprecipitation (IP), and GAPDH (loading control) panels are indicated. Unless noted otherwise (Figs. 1D, G, 2B, 4A, D, F and S1A, S4B, S5A, B and S7C, D), samples were derived from the same experiment and processed on the same gel. Representative immunoblots and microscopy images were reproduced in at least two independent experiments with similar results.

## Statistical analysis

Statistical analyses were performed using GraphPad Prism 10. Data are presented as mean ± SD from at least three independent biological replicates unless indicated otherwise. Comparisons between two groups were evaluated using two-sided unpaired Student's *t* tests, and multiple-group comparisons were analyzed by one-way ANOVA followed by Tukey's multiple-comparison test. Statistical significance was defined as $P < 0.05$. No data were excluded, and analyses were conducted without randomization or blinding unless specified. The number of independent replicates, the number analyzed cells per replicate, normalization methods, and statistical tests are detailed in the figure legends.

## Reporting summary

Further information on research design is available in the Nature Portfolio Reporting Summary linked to this article.

## Data availability

All data are available within the article and its Supplementary Information. Source data are provided with this paper.

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

## Acknowledgements

We thank Tadashi Yamamoto and Tomomi Kiyomitsu for sharing reagents; the OIST Animal Resource Section for support; Muhammad Hamzah for feedback on the manuscript, and members of the Cell Proliferation and Gene Editing Unit for discussion. This work was supported by the Okinawa Institute of Science and Technology and the Japan Society for the Promotion of Science [KAKENHI, 23K05773 and 25H02402 to F.M. and 24K09461 to M.O.].

## Author contributions

Conceptualization: H.B., M.O., F.M.; funding acquisition: F.M., M.O.; investigation: H.B., F.M.; methodology: H.B., E.F.Y.N., M.O., F.M.; resources: F.M.; writing—original draft: H.B., F.M.; writing—review & editing: H.B., E.F.Y.N., M.O., F.M.

## Competing interests

The authors declare no competing interests
