## [Transparent Peer Review file · Nature Communications]

Cancer-Associated USP28 Missense Mutations Disrupt 53BP1 Interaction and p53 Stabilization

Corresponding Author: Professor Franz Meitinger

Version 0:

Reviewer comments:

Reviewer #1

(Remarks to the Author)
Referee Report

Title: Cancer-Associated USP28 Missense Mutations Disrupt 53BP1 Interaction and p53 Stabilization
Authors: Hazrat Belal et al.

Summary

This manuscript investigates how cancer-associated missense mutations in the deubiquitinase USP28 compromise its interaction with 53BP1 and impair p53 stabilization in response to mitotic stress. The authors convincingly show that a specific USP28 isoform (USP28hF2) mediates this function through its C-terminal domain and dimerization capacity. Cancer-derived mutations in this region result in protein destabilization, nuclear exclusion, or loss of 53BP1 binding, thereby reducing the cell's ability to activate p53 after mitotic arrest.

The manuscript addresses an important question in mitotic surveillance and tumor suppression and is based on well-designed experiments with broad implications. The synthetic dimerization strategy is particularly elegant. However, several key controls are missing, and some mechanistic claims (especially involving PLK1) require further validation or moderation. With these revisions, the study will make a strong contribution.

Major Points

Lack of asynchronous controls in co-IP experiments (Figs. 1F, 1H, 2E, 2F, 4, 6D)

The authors do not include immunoprecipitation controls from asynchronous cells, making it difficult to assess whether the observed complexes are specifically induced by mitotic stress. Without this control, the data do not unambiguously support the conclusion that these are mitotic "stopwatch complexes." Repeating key co-IPs with asynchronous extracts on the same gel is necessary to validate the stress-induced nature of these interactions.

Missing WT control in Fig. 3A

The figure evaluates USP28 truncation mutants, but does not show a WT control expressed under the same conditions. This control is essential to interpret the impact of deletions on 53BP1 binding and p53 stabilization and must be included.

PLK1 dependency should be revised or better supported (lines 270–279)

The claim that PLK1 activity is required for the interaction between USP28 and 53BP1 is insufficiently supported. As shown by Burigotto et al., EMBO Reports 2023, PLK1 inhibition causes 53BP1 to remain kinetochore-bound, potentially preventing complex formation. This alternative explanation must be acknowledged. The authors should either remove the PLK1 section (including Figure 4) or provide additional evidence, such as using the 53BP1 phosphorylation-deficient mutant described in Meitinger et al., Science 2024.

Ambiguity in mutation distribution (Fig. 2B / S3A)

The presentation of clone-specific mutations in Figure 2B is unclear. Supplementary Figure S3A provides more detailed information and would be better placed in the main figure panel to aid interpretation.

Positive note: synthetic dimerization experiments

The experiments involving chemically induced dimerization of USP28 truncations are technically sophisticated and particularly compelling. They effectively demonstrate the necessity of dimerization for 53BP1 binding and add strength to the mechanistic conclusions.

Minor Points

Duplicate content in lines 77–87

There appears to be a duplication of text in this paragraph. Please revise to eliminate redundancy.

Overstatement in lines 373–374

The sentence “we elucidate the molecular mechanism” may overstate the conclusions. While the study defines important structural domains and interactions, the precise enzymatic mechanism by which USP28 regulates p53 ubiquitination is not yet resolved and remains a key open question.

RT-PCR isoform analysis (Fig. 1L)

The RT-PCR results suggest that the shorter USP28 isoform is the predominant transcript in multiple cancer cell lines. However, conventional PCR may preferentially amplify the shorter product, especially in the presence of both isoforms. To address this, the authors should perform control PCRs using known plasmid mixes (e.g., 50:50, 90:10, 10:90 ratios) to verify that the assay does not mask co-expression of the longer isoform. This is particularly important given the biological implications of isoform-specific function.

Recommendation:

Minor revision

The manuscript presents strong and novel findings of broad interest to cancer and cell cycle biology. Addressing the missing controls and moderating certain mechanistic claims will significantly enhance the clarity and impact of the study. With these revisions, I would support publication in Nature Communications.

Reviewer #2

(Remarks to the Author)

USP28 and 53BP1 play key, poorly understood roles in the activation of p53 in response to mitotic stress. Here, Belal et al examine the mechanisms in further detail. Using a competition assay with mitotic delays or DNA damage, they find that deleting USP28 or 53BP1 reduced the sensitivity of cancer cells to mitotic delays, while having only a minor effect following DNA damage. In p53 proficient cells, they observed lower p53 levels and reduced responses to mitotic stress and damage but no effect on Myc. They next examined 2 isoforms of USP28 and found that IF2, a shorter isoform, interacted with 53BP1 and that structural predictions suggested that the additional sequence in IF1 would block the 53BP1 interaction. Analysis of a large number of cell lines and tissues only found IF1 expression in mouse brain tissues or a mouse brain tumor line, indicating that IF2 is the predominant isoform. They then demonstrated via live cell imaging that only IF2 can efficiently activate p53 responses following monastrol induced mitotic delays. Following the generation of complemented RPE1 cells, they found mutations in USP28 in around half of their clones. Testing these, they identified mutations in both the USP and C-terminus that led to reduced sensitivity to mitotic delay, as well as several that impaired nuclear localization apart from the NLS. Focusing on a frameshift mutant, they found that it truncated the C-terminus and they mapped the last 13 aa as key for the interactions with 53BP1 and p53 activation. Surprisingly, they found that this did not interact on its own and required the dimerization of USP28 to be a functional binding surface using an inducible dimer of USP28 and dimerization mutants. Moreover, this interaction was PLK1 dependent and independent of the UBA and UIM domains identified in the N-terminus of USP28. They next examine cancer associated missense mutations in USP28 and identified mutations in the NLS and C-terminal domain, including mutations that affected stability and nuclear localization from outside of the NLS. Functional analysis of a subset of these, as well as mutations in certain cancer cell lines, revealed that they all impaired the mitotic stopwatch complex.

The manuscript is very systematic and easy to follow, despite having some very complex experiments. It very nicely breaks down the molecular requirements for complex formation and function and places them in the context of cancer where USP28 is known to be mutated. The figures are very nice and the data appears to be of high quality, although little/no primary data is provided in some cases. I have a few minor comments to consider addressing but overall think this is a very nice manuscript that will be of strong interest to a broad readership.

Some specific comments:

1. Line 77: I would recommend writing cancer-ass as cancer-associated.
2. In Figure 1E, the authors demonstrate USP28-dependent sensitivity of RPE1 cells to mitotic stress using live-cell imaging. Including representative images or movies in the figure would help visualize the observed phenotypes.
3. In Figure 1D, quantification of the western blots is shown below the blots. However, presenting the data as bar graphs would improve clarity and make the differences in protein levels easier to interpret.
4. In Figure 4B, The 53BP1-p53 interaction is ok in non dimerized conditions and then in the dimer, PLK1i causes both interactions to be lost, indicating that 53BP1 interacts with p53 in a PLK1 dep manner regardless of USP28. USP28 dimerization then allows its interaction- what controls it? Is this ATM dependent?
5. In Figure 5G and I, only one cell is shown in the images, it would be nice to have some representative fields where groups of cells are shown and add quantification.
6. In Figure 3I, why is the middle blot cut and the top and bottom blots that are supposed to be comparable not cut?
7. There is a lot of redundancy between the last two paragraphs of Introduction (page 4), that could be consolidated.
8. I was surprised that mutations in the NLS were the most frequently observed. Is there a potential benefit to cancers maintaining active cytoplasmic USP28?
9. Are any mutations associated with clinical data that would determine if the use of mitotic agents is associated with the selection of these mutations?

Reviewer #3

(Remarks to the Author)

In the manuscript "Cancer-Associated USP28 Missense Mutations Disrupt 53BP1 Interaction and p53 Stabilization", Meitinger and coworkers analyze the interaction of the Usp28 deubiquitinase and the key DDR protein 53bp1 and explore the role of this interaction in regulation of p53 levels. They identify the region in the Usp28 that mediates the interaction with 53bp1 and examine the impact of mutations in Usp28 on subcellular localization of Usp28, 53bp1 binding and stabilization of p53.

The data are of interest to the scientists working in the field and may provide important insights for therapeutic exploitation of the pathway. The experiments are well controlled and largely support the authors' conclusions. The data are of high quality and well presented and the manuscript is well written. However, I believe that the authors should perform additional experiments to support the main conclusions of the study and enhance their dataset. Furthermore, occasionally the data are overinterpreted and statements based on the published data are not inclusive, so the authors need to adjust the phrasing or support their arguments with new experiments, as detailed below.

1. The authors analyze a large number of Usp28 variants that either don't bind 53bp1 or don't dimerize and show effects on p53 protein levels. I think it is crucial to show that these mutations impact the ability of Usp28 to deubiquitinate p53 in vitro and in cells. Such ubiquitination assays are well established and would greatly strengthen the dataset.
2. Along these lines, the authors argue that Usp28 leads to stabilization of p53 but no experiment that directly measures protein half life is shown. I realize that p53 protein levels under stress are mainly regulated via protein turnover, but it would be important to see a cycloheximide chase or a similar assay to compare the effect of wildtype Usp28 and some of the key variants. The authors do this for Myc so it should be rather straightforward to perform these experiments for p53.
3. Fig 2d shows that the C171 mutant is better than Usp28-KO in increasing p53 levels, whereas some other mutations have a comparable/stronger impact. Does this imply that Usp28 can regulate p53 independent of its deubiquitinase activity? The ubiquitination assays suggested above could also help to address this point.
4. The authors use WB and IF data to argue for stabilization of p53. At least visually these experiments do not always agree. For example, in Fig. 5H and 5I, the WB shows a small decrease in p53 levels for the R141C mutant, whereas in the IF experiment, p53 is completely gone. Quantification of both types of assay in every analysis would be helpful.
5. The authors find a number of mutations in the shorter isoform of Usp28. Are mutations also found in the longer hIF1 isoform, which doesn't bind 53bp1?
6. The manuscript would strongly benefit from additional functional data documenting the biological effects of different Usp28 variants under mitotic stress. Authors could analyze p53-dependent gene expression, compare cell survival with and without mitotic poisons; ideally - compare tumor development in animal models for wildtype Usp28 and a cancer-associated Usp28 variant that impairs 53bp1 interaction.
7. The model is that Plk1-mediated phosphorylation promotes Usp28-53bp1 interaction. Which protein is exactly phosphorylated - I found mixed sentences in the text. Has the phosphosite been mapped and mutated to validate this model? If such mutations exist, they should be used in parallel with chemical inhibition of PLKi.
8. I don't understand why the Usp28-53bp1 complex in the G1 phase cells should necessarily be formed in mitosis. Even if PLK1 is completely inactive in G1, I think it's unlikely that all of Usp28 or 53bp1 that is phosphorylated in mitosis would be degraded or dephosphorylated before G1. Furthermore, the data shown for the R141C mutant in (Fig. 5E-I) suggest that the Usp28-53bp1 complex is formed but is not transmitted to the daughter cells since both proteins show a completely non-overlapping localization pattern in G1.
9. I think Fig 3f shows that the dimer deficient variant Usp28-VLEE increases p53 levels in mitotic cells, despite a defect in 53bp1 interaction - this appears to contradict the authors' model.
10. The IF images shown in Fig. 5g,i,k and others should be quantified as done for some IF data (eg, Fig 2d).
11. The last paragraph in the introduction is repeated twice.

Minor comments:

1. On page the authors write: "Although PLK1 is active only during mitosis...". This statement contradicts many studies showing S phase effects of Plk1 on DNA replication, etc. and should be rephrased.
2. What is the measure of confidence of the AlphaFold model that would support the following statement on page 6: "Structural predictions from AlphaFold with high-confidence modeling indicate that exon 19 forms an additional alpha helix in isoform 1, which may obstruct the interaction surface for 53BP1 (Fig. 1I; S2C)."

3. On page 7 the authors write: "Since USP28 overexpression is toxic, we reasoned that all the observed mutations specifically impair p53 stabilization (Fig. 2B; S3A) .

Overexpression of Usp28 is not generally toxic to tumor cells, as was shown by multiple previous studies. Perhaps it's true in the context of mitotic stress, but this should be stated explicitly.

4. I don't understand the meaning of the following passage on page 5 - "In contrast, USP28 Δ had a less pronounced effect, and TP53BP1 Δ had no effect on the sensitivity to DXR-induced DNA damage. The later observation could be explained by the independent role of 53BP1 in DNA repair."

First, it's surprising that 53bp1 loss doesn't alter sensitivity to DXR. Second, this result rather suggests a 53bp1-independent role of Usp28 in sensitivity to DXR-induced DNA damage. This should be rephrased.

5. I also don't understand the following argument on page 7, "Notably, the expression levels of all tested mutants, except G903A and P953L, were 5- to 20-fold higher than in wildtype RPE1 cells (Fig. 2D; S3B, C), suggesting that the mutants fail to stabilize p53 following an extended mitotic duration even though the expression level is significantly increased."

This assumes that p53 is the only Usp28 effector, which is not true. Rephrasing this sentence can help.

6. On page 12, the authors write: "Using transgene expression, we found that the tumor suppressor function of USP28 is mediated by the shorter isoform 2 of USP28..."

As the authors do now directly analyze any effect on tumor suppression, this should be rephrased.

Reviewer #4

(Remarks to the Author)

Version 1:

Reviewer comments:

Reviewer #1

(Remarks to the Author)

All my comments have been addressed satisfactorily, and I am pleased with the thorough and thoughtful revisions. I would like to congratulate the Meitinger lab on this impressive first independent study.

Reviewer #2

(Remarks to the Author)

The authors have addressed all the major critiques of the reviewers and I do not have any further comments.

Reviewer #3

(Remarks to the Author)

In the revised version of the manuscript "Cancer-Associated USP28 Missense Mutations Disrupt 53BP1 Interaction and p53 Stabilization", Meitinger and co-workers have addressed most of the concerns raised in the original review. I don't have further questions. I believe it is an important and well-controlled study and should be interesting to many scientists in the field.

Reviewer #4

(Remarks to the Author)

We sincerely thank the reviewers for their constructive and positive feedback and insightful comments, which have greatly improved the quality of our manuscript.

REVIEWER COMMENTS

Reviewer #1 (Remarks to the Author):

Referee Report

Title: Cancer-Associated USP28 Missense Mutations Disrupt 53BP1 Interaction and p53 Stabilization

Authors: Hazrat Belal et al.

Summary

This manuscript investigates how cancer-associated missense mutations in the deubiquitinase USP28 compromise its interaction with 53BP1 and impair p53 stabilization in response to mitotic stress. The authors convincingly show that a specific USP28 isoform (USP28hIF2) mediates this function through its C-terminal domain and dimerization capacity. Cancer-derived mutations in this region result in protein destabilization, nuclear exclusion, or loss of 53BP1 binding, thereby reducing the cell's ability to activate p53 after mitotic arrest.

The manuscript addresses an important question in mitotic surveillance and tumor suppression and is based on well-designed experiments with broad implications. The synthetic dimerization strategy is particularly elegant. However, several key controls are missing, and some mechanistic claims (especially involving PLK1) require further validation or moderation. With these revisions, the study will make a strong contribution.

Major Points

Lack of asynchronous controls in co-IP experiments (Figs. 1F, 1H, 2E, 2F, 4, 6D)

The authors do not include immunoprecipitation controls from asynchronous cells, making it difficult to assess whether the observed complexes are specifically induced by mitotic stress. Without this control, the data do not unambiguously support the conclusion that these are mitotic "stopwatch complexes." Repeating key co-IPs with asynchronous extracts on the same gel is necessary to validate the stress-induced nature of these interactions.

We agree with the reviewer that it is important to assess whether the complex forms specifically during prolonged mitosis. We addressed this question extensively in our previous study (PMID: 38547292), where we demonstrated that the complex forms specifically in mitotically arrested cells. No or significantly less complex formation was observed in cells arrested in G1 or G2 phase (PMID: 38547292 Fig. 2C), in cells with induced DNA damage (PMID: 38547292 Fig. 2D), or in normal mitotic cells (Fig. 2H). Furthermore, we showed that the complex depends on the mitotic kinase PLK1, which is predominantly active during mitosis (PMID: 38547292 Fig. 3). These data support that the soluble mitotic stopwatch complex forms specifically during mitosis.

To further strengthen this point in our current manuscript, we now provide additional data, including asynchronous cell controls. These controls are shown in Figures 1H, 2B (corresponding to the experiment previously shown in 1F), and 2D (corresponding to the experiment previously shown in 1H). To further emphasize the mitosis-specific nature of this complex, we now show in Figure 1I that its formation depends on PLK1 (previously shown in Fig. 4A).

Missing WT control in Fig. 3A

The figure evaluates USP28 truncation mutants, but does not show a WT control expressed under the same conditions. This control is essential to interpret the impact of deletions on 53BP1 binding and p53 stabilization and must be included.

We repeated the experiment and included the wildtype (WT) control expressing the USP28 hIF2 WT transgene, now shown in Figure 4A. Figure 4A demonstrates that the mutant lacking the C-terminal 13 amino acids fails to interact with 53BP1, whereas the WT transgene retains this interaction. In contrast to the WT transgene, the deletion mutant also fails to stabilize p53 following mitotic exit in G1 phase.

PLK1 dependency should be revised or better supported (lines 270–279)

The claim that PLK1 activity is required for the interaction between USP28 and 53BP1 is insufficiently supported. As shown by Burigotto et al., EMBO Reports 2023, PLK1 inhibition causes 53BP1 to remain kinetochore-bound, potentially preventing complex formation. This alternative explanation must be acknowledged. The authors should either remove the PLK1 section (including Figure 4) or provide additional evidence, such as using the 53BP1 phosphorylation-deficient mutant described in Meitinger et al., Science 2024.

We have previously shown that the interaction between 53BP1 and USP28 depends on PLK1 (PMID: 38547292). Burigotto et al. investigated the interaction between 53BP1 and p53; however, to the best of our knowledge, their study does not provide information regarding the interaction between USP28 and 53BP1. However, we acknowledge the alternative possibility suggested by Burigotto et al. and cite this paper (PMID: 37888778; Ref 34).

Focusing on our own data, we found that the interaction between USP28 and 53BP1 requires both the C-terminal domain of USP28 (amino acids 651–1045) and its dimerization domain (Figure 4). This suggests two potential roles for PLK1: either PLK1 promotes USP28 dimerization, or it facilitates the interaction between the USP28 C-terminus and 53BP1. Our experiments support the latter. Specifically, when we enforce dimerization in a PLK1-independent manner, the USP28-53BP1 interaction still depends on PLK1. This indicates that PLK1 promotes the interaction between the USP28 C-terminus and 53BP1 — an important finding. These data are presented in Figure S6 (before Figure 4). Furthermore, our results show that the interaction between the C-terminus of USP28 and 53BP1 remains stable after mitosis, mirroring the behavior of full-length USP28.

To identify the relevant target of PLK1, we mutated 16 putative PLK1 phosphorylation sites predicted using the GPS 6.0 software. This mutant still retained its ability to interact with 53BP1, suggesting that USP28 is not the direct target of PLK1. Although we have previously mapped potential phosphorylation sites on 53BP1, we have not yet identified the site relevant for USP28 binding (PMID: 38547292). Therefore, at this point, we cannot pinpoint the direct target of PLK1 in this process. However, the central claim that PLK1 promotes the interaction between the C-terminus of USP28 and 53BP1 remains true.

As our previous manuscript did not clearly outline the two possible mechanisms by which PLK1 could control the USP28–53BP1 interaction (dimerization vs. direct interaction), we have revised the text accordingly. Since we cannot provide direct evidence for the phosphorylation sites, we have moved the corresponding data to Supplementary Figure S6 and acknowledged the alternative mechanism proposed by Burigotto et al., which is now cited in the revised manuscript (PMID: 37888778; Ref 34).

Ambiguity in mutation distribution (Fig. 2B / S3A)

The presentation of clone-specific mutations in Figure 2B is unclear. Supplementary Figure S3A provides more detailed information and would be better placed in the main figure panel to aid interpretation.

We prefer to keep the detailed figure showing the distribution of individual mutations in the supplementary material to maintain clarity and simplicity in the main figures. The simplified figure in the main text is intended to provide a clear overview of the domains in which the mutations occur, without overwhelming the reader with excessive detail. The more detailed information is still available for the interested reader in Figure S3A.

Positive note: synthetic dimerization experiments

The experiments involving chemically induced dimerization of USP28 truncations are technically sophisticated and particularly compelling. They effectively demonstrate the necessity of dimerization for 53BP1 binding and add strength to the mechanistic conclusions.

We appreciate the reviewer's positive feedback.

Minor Points

Duplicate content in lines 77–87

There appears to be a duplication of text in this paragraph. Please revise to eliminate redundancy.

We deleted the duplicated paragraph.

Overstatement in lines 373–374

The sentence “we elucidate the molecular mechanism” may overstate the conclusions. While the study defines important structural domains and interactions, the precise enzymatic

mechanism by which USP28 regulates p53 ubiquitination is not yet resolved and remains a key open question.

We toned down the statement.

Before: Furthermore, we elucidate the molecular mechanisms by which USP28 stabilizes p53 following mitotic stress (Fig. 2-5) and show that these mechanisms are frequently inactivated in cancer (Fig. 6, 7).

Now: Furthermore, we identified molecular mechanisms of USP28 that are required for p53 stabilization following mitotic stress (Fig. 2-4) and demonstrate that these mechanisms are frequently disrupted in cancer (Fig. 5-7).

RT-PCR isoform analysis (Fig. 1L)

The RT-PCR results suggest that the shorter USP28 isoform is the predominant transcript in multiple cancer cell lines. However, conventional PCR may preferentially amplify the shorter product, especially in the presence of both isoforms. To address this, the authors should perform control PCRs using known plasmid mixes (e.g., 50:50, 90:10, 10:90 ratios) to verify that the assay does not mask co-expression of the longer isoform. This is particularly important given the biological implications of isoform-specific function.

According to the reviewer's suggestion, we performed the requested control PCRs using known plasmid mixes (50:50, 90:10, and 10:90 ratios). We found that even 10% of the short or long isoforms can be reliably detected with the method used. The corresponding data are shown in Figure S2E.

Recommendation:

Minor revision

The manuscript presents strong and novel findings of broad interest to cancer and cell cycle biology. Addressing the missing controls and moderating certain mechanistic claims will significantly enhance the clarity and impact of the study. With these revisions, I would support publication in Nature Communications.

We thank the reviewer for the positive evaluation and constructive feedback on our manuscript.

Reviewer #2 (Remarks to the Author):

USP28 and 53BP1 play key, poorly understood roles in the activation of p53 in response to mitotic stress. Here, Belal et al examine the mechanisms in further detail. Using a competition assay with mitotic delays or DNA damage, they find that deleting USP28 or 53BP1 reduced the sensitivity of cancer cells to mitotic delays, while having only a minor effect following DNA damage. In p53 proficient cells, they observed lower p53 levels and reduced responses to

mitotic stress and damage but no effect on Myc. They next examined 2 isoforms of USP28 and found that IF2, a shorter isoform, interacted with 53BP1 and that structural predictions suggested that the additional sequence in IF1 would block the 53BP1 interaction. Analysis of a large number of cell lines and tissues only found IF1 expression in mouse brain tissues or a mouse brain tumor line, indicating that IF2 is the predominant isoform. They then demonstrated via live cell imaging that only IF2 can efficiently activate p53 responses following monastrol induced mitotic delays. Following the generation of complemented RPE1 cells, they found mutations in USP28 in around half of their clones. Testing these, they identified mutations in both the USP and C-terminus that led to reduced sensitivity to mitotic delay, as well as several that impaired nuclear localization apart from the NLS. Focusing on a frameshift mutant, they found that it truncated the C-terminus and they mapped the last 13 aa as key for the interactions with 53BP1 and p53 activation. Surprisingly, they found that this did not interact on its own and required the dimerization of USP28 to be a functional binding surface using an inducible dimer of USP28 and dimerization mutants. Moreover, this interaction was PLK1 dependent and independent of the UBA and UIM domains identified in the N-terminus of USP28. They next examine cancer associated missense mutations in USP28 and identified mutations in the NLS and C-terminal domain, including mutations that affected stability and nuclear localization from outside of the NLS. Functional analysis of a subset of these, as well as mutations in certain cancer cell lines, revealed that they all impaired the mitotic stopwatch complex.

The manuscript is very systematic and easy to follow, despite having some very complex experiments. It very nicely breaks down the molecular requirements for complex formation and function and places them in the context of cancer where USP28 is known to be mutated. The figures are very nice and the data appears to be of high quality, although little/no primary data is provided in some cases. I have a few minor comments to consider addressing but overall think this is a very nice manuscript that will be of strong interest to a broad readership.

Some specific comments:

1. Line 77: I would recommend writing cancer-ass as cancer-associated.

We deleted the entire paragraph as this contained redundant information.

2. In Figure 1E, the authors demonstrate USP28-dependent sensitivity of RPE1 cells to mitotic stress using live-cell imaging. Including representative images or movies in the figure would help visualize the observed phenotypes.

We included representative images which are now shown in Figure 2A and S1H.

3. In Figure 1D, quantification of the western blots is shown below the blots. However, presenting the data as bar graphs would improve clarity and make the differences in protein levels easier to interpret.

We included bar graphs as suggested (Figure 1D).

4. In Figure 4B, The 53BP1-p53 interaction is ok in non dimerized conditions and then in the dimer, PLK1i causes both interactions to be lost, indicating that 53BP1 interacts with p53 in a PLK1 dep manner regardless of USP28. USP28 dimerization then allows its interaction- what controls it? Is this ATM dependent?

We agree with the reviewer that the 53BP1–p53 interaction is PLK1-dependent regardless of USP28 status. In our previous work (PMID: 38547292), we mapped the PLK1 phosphorylation sites (T1756, S1758, S1759) on 53BP1 that are required for p53 binding. For the 53BP1–USP28 interaction, the exact PLK1 target site remains unknown. Our data show that the C-terminal domain of USP28 (residues 651–1045) mediates 53BP1 binding, and that this interaction is PLK1-dependent but stable after mitosis.

USP28 dimerization appears to be PLK1-independent. Consistent with published work (PMID: 30926243; PMID: 30926242), USP28 predominantly exists as a dimer, and ATM can promote monomerization after DNA damage — a process unrelated to mitotic stress (PMID: 38227944). We previously showed that the mitotic stopwatch is independent of ATM (PMID: 38547292). We therefore consider it unlikely that ATM controls the 53BP1–USP28 interaction. To test whether USP28 itself is a PLK1 substrate, we generated a mutant in which 16 predicted PLK1 sites were replaced with alanine. Despite reduced expression, this mutant still bound 53BP1, suggesting that PLK1 does not directly phosphorylate USP28 for the interaction.

5. In Figure 5G and I, only one cell is shown in the images, it would be nice to have some representative fields where groups of cells are shown and add quantification.

We added a bigger field with a group of cells and quantifications. See also Point 10 from Reviewer 3.

6. In Figure 3I, why is the middle blot cut and the top and bottom blots that are supposed to be comparable not cut?

The expression of the USP28 (399–1045) construct is weaker compared to the others. For visualization purposes, we present two different exposures in Figure 4I (previously Figure 3I). The uncropped original blots with both low and high exposures are provided in Figure S5F (previously Figure S4F). To address the reviewer’s concern and avoid confusion, we added a description to the figure legend of Figure 4I.

The key comparison is between the same constructs with and without the dimerizer. Unlike the shorter fragment (651–1045), the longer fragment (399–1045), which includes the dimerization arm, does not require the chemical dimerizer to mediate the interaction.

7. There is a lot of redundancy between the last two paragraphs of Introduction (page 4), that could be consolidated.

We removed the duplicated paragraph.

8. I was surprised that mutations in the NLS were the most frequently observed. Is there a potential benefit to cancers maintaining active cytoplasmic USP28?

This is an interesting question. One possibility is that cytoplasmic USP28 stabilizes other substrates that are essential for cancer growth. Identifying these substrates will be an important question to address in future studies.

9. Are any mutations associated with clinical data that would determine if the use of mitotic agents is associated with the selection of these mutations?

Thank you for this insightful comment. We explored available clinical datasets to investigate whether the identified mutations are associated with treatment history involving mitotic agents. However, we were unable to find data that directly links the presence or selection of these mutations to the use of mitotic agents. We agree that such information would be highly valuable, and we anticipate that future studies integrating genomic and detailed treatment data will help address this important question.

Reviewer #3 (Remarks to the Author):

In the manuscript “Cancer-Associated USP28 Missense Mutations Disrupt 53BP1 Interaction and p53 Stabilization”, Meitinger and coworkers analyze the interaction of the Usp28 deubiquitinase and the key DDR protein 53bp1 and explore the role of this interaction in regulation of p53 levels. They identify the region in the Usp28 that mediates the interaction with 53bp1 and examine the impact of mutations in Usp28 on subcellular localization of Usp28, 53bp1 binding and stabilization of p53.

The data are of interest to the scientists working in the field and may provide important insights for therapeutic exploitation of the pathway. The experiments are well controlled and largely support the authors’ conclusions. The data are of high quality and well presented and the manuscript is well written. However, I believe that the authors should perform additional experiments to support the main conclusions of the study and enhance their dataset. Furthermore, occasionally the data are overinterpreted and statements based on the published data are not inclusive, so the authors need to adjust the phrasing or support their arguments with new experiments, as detailed below.

1. The authors analyze a large number of Usp28 variants that either don’t bind 53bp1 or don’t dimerize and show effects on p53 protein levels. I think it is crucial to show that these mutations impact the ability of Usp28 to deubiquitinate p53 in vitro and in cells. Such ubiquitination assays are well established and would greatly strengthen the dataset.

Thank you for the important suggestion. Our data show that p53 levels increase in response to mitotic stress, which could arise from changes in transcription, translation, or protein degradation. To distinguish these possibilities, we performed cycloheximide (CHX) chase assays to block translation. In RPE1 wild-type cells, p53 levels declined during CHX treatment, indicating that p53 abundance is primarily regulated via protein degradation (Fig. 1E, F). Consistent with previous reports (PMID: 27432896), USP28 knockout cells showed a similar p53 degradation rate under basal conditions.

Upon mitotic stress induced by PLK4 inhibition, the p53 half-life in wild-type cells increased significantly, whereas in USP28 knockout cells it remained unchanged, indicating that USP28 specifically stabilizes p53 following mitotic stress (Fig. 1E, F). Likewise, mutations in the USP28 deubiquitinase domain and C-terminal truncations that disrupt 53BP1 interaction phenocopied USP28 loss (Fig. 3G, S4).

To directly assess USP28's role in deubiquitinating p53, we immunoprecipitated p53 under denaturing conditions after mitotic stress. We observed a marked accumulation of ubiquitylated p53 in USP28 knockout or USP28 mutant cells compared to wild-type, supporting that USP28 prevents ubiquitin-mediated p53 degradation (Fig. 1G, S4B, C) (PMID: 27371829). These ubiquitination assays thus confirm that the mutations impair USP28's deubiquitinase activity toward p53 both in vitro and in cells, strengthening our mechanistic conclusions.

2. Along these lines, the authors argue that Usp28 leads to stabilization of p53 but no experiment that directly measures protein half life is shown. I realize that p53 protein levels under stress are mainly regulated via protein turnover, but it would be important to see a cycloheximide chase or a similar assay to compare the effect of wildtype Usp28 and some of the key variants. The authors do this for Myc so it should be rather straightforward to perform these experiments for p53.

We measured the half-life of p53 in USP28 Δ cells and various USP28 mutants and found that p53 half-life is markedly reduced in these mutants following mitotic stress (Figure 1E, F; 3G; S4A). For a more detailed explanation, please refer to the previous point.

3. Fig 2d shows that the C171 mutant is better than Usp28-KO in increasing p53 levels, whereas some other mutations have a comparable/stronger impact. Does this imply that Usp28 can regulate p53 independent of its deubiquitinase activity? The ubiquitination assays suggested above could also help to address this point.

We agree that in the single-cell tracking experiment (Figure 3C), the C171A mutant appears less severe than some other mutants. However, it shows a comparable effect on p53 stabilization in bulk measurements (Figure 3E, F). Our deubiquitination and cycloheximide assays indicate that C171A behaves as a complete loss-of-function mutant, similar to USP28 Δ (compare Figure 3G and S4A with Figure 1E, F; and Figure S4B, C with Figure 1G). Based on these results, we conclude that the C171A mutation impairs p53 stabilization following mitotic stress and does not support p53 regulation independent of USP28's deubiquitinase activity.

4. The authors use WB and IF data to argue for stabilization of p53. At least visually these

experiments do not always agree. For example, in Fig. 5H and 5I, the WB shows a small decrease in p53 levels for the R141C mutant, whereas in the IF experiment, p53 is completely gone. Quantification of both types of assay in every analysis would be helpful.

Thank you for the helpful suggestion. We have now included quantifications for both the Western blot (Figure 5H) and immunofluorescence experiments (Figure 5I). These analyses confirm and complement each other, providing a clearer and more comprehensive picture of p53 stabilization across different mutants.

5. The authors find a number of mutations in the shorter isoform of Usp28. Are mutations also found in the longer hIF1 isoform, which doesn't bind 53bp1?

The longer isoform is the same gene which expresses an additional exon 19. This exon codes for 30 amino acids. In this area six mutations have been found according to the COSMIC database. The mutation rate in exon 19 is similar to the N-terminus (aa1-157), which is not required for mitotic stopwatch function (Figure 5B, C; S7).

6. The manuscript would strongly benefit from additional functional data documenting the biological effects of different Usp28 variants under mitotic stress. Authors could analyze p53-dependent gene expression, compare cell survival with and without mitotic poisons; ideally - compare tumor development in animal models for wildtype Usp28 and a cancer-associated Usp28 variant that impairs 53bp1 interaction.

Thank you for this valuable suggestion. In a prior genome-wide CRISPR-Cas9 screen, we identified USP28, 53BP1, p21, and p53 as the principal regulators of cell proliferation in response to mitotic stress in RPE1 cells (PMID: 27432897). These were the only genes whose loss significantly affected proliferation under these conditions, a finding corroborated by independent studies (PMID: 27432896; PMID: 27371829). Given this strong genetic evidence, we believe that additional p53-dependent gene expression profiling would be unlikely to yield novel insights beyond our current findings.

Instead, we have systematically characterized the functional impact of specific USP28 mutations using single-cell tracking to measure mitotic durations and analyzed the effects of PLK4-induced mitotic stress on both wild-type and mutant cells. These complementary approaches consistently demonstrate that mitotic stress induces a USP28-dependent cell cycle arrest, which is compromised in the mutants examined.

We agree that in vivo validation, including tumor development assays with cancer-associated USP28 variants, would be valuable. However, such studies are beyond the scope of this mechanistic, in vitro-focused investigation. We consider this an important direction for future work to establish the physiological relevance of our findings.

7. The model is that Plk1-mediated phosphorylation promotes Usp28-53bp1 interaction. Which protein is exactly phosphorylated - I found mixed sentences in the text. Has the phosphosite been mapped and mutated to validate this model? If such mutations exist, they should be used

in parallel with chemical inhibition of PLKi.

To identify the relevant target of PLK1, we mutated 16 putative PLK1 phosphorylation sites predicted using the GPS 6.0 software. This mutant still retained its ability to interact with 53BP1, suggesting that USP28 is not the direct target of PLK1. Although we have previously mapped potential phosphorylation sites on 53BP1, we have not yet identified the site relevant for USP28 binding (PMID: 38547292). Therefore, at this point, we cannot pinpoint the direct target of PLK1 in this process. However, the central claim that PLK1 promotes the interaction between the C-terminus of USP28 and 53BP1 remains true.

Since we haven't clearly outlined the potential two potential possibilities of PLK1 in controlling USP28-53BP1 interaction in our previous manuscript (Dimerization vs. Interaction), we revised the text accordingly. Since we cannot provide the direct phosphorylation sites, we moved the data to Supplementary Figure S6 and acknowledge the alternative possibility suggested by Burigotto et al. and cite this paper (PMID: 37888778; Ref 34).

8. I don't understand why the Usp28-53bp1 complex in the G1 phase cells should necessarily be formed in mitosis. Even if PLK1 is completely inactive in G1, I think it's unlikely that all of Usp28 or 53bp1 that is phosphorylated in mitosis would be degraded or dephosphorylated before G1. Furthermore, the data shown for the R141C mutant in (Fig. 5E-I) suggest that the Usp28-53bp1 complex is formed but is not transmitted to the daughter cells since both proteins show a completely non-overlapping localization pattern in G1.

We believe that USP28–53BP1 complexes present in G1 cells are assembled during mitosis, consistent with their role in measuring mitotic duration. In our previous work (PMID: 38547292), we showed that prolonged mitosis increases the number of USP28–53BP1 complexes formed. How the stability of this complex is maintained remains an open question. We considered two possibilities: (1) the Mitotic Stopwatch Complex forms in mitosis and remains stable after mitotic exit, or (2) mitotic phosphorylations that persist into G1 promote complex formation. The prevailing model is that PLK1-dependent phosphorylations are rapidly removed after mitosis (PMID: 21750572). For 53BP1, two sites in the UDR domain have been characterized (PMID: 24703952), and mitotic phosphorylation at these sites is quickly removed to allow DNA binding in interphase. For one site (S1618), we confirmed using a phospho-specific antibody that phosphorylation is PLK1-dependent and absent 6 h after release from mitotic arrest, matching our experimental timing.

These data support a model in which the USP28–53BP1 complex forms during mitosis in a PLK1-dependent manner and remains stable after mitotic exit. However, we cannot rule out that some phosphorylations persist beyond mitosis and help maintain the complex in G1. We now acknowledge that the interaction may be more dynamic than previously appreciated (Page 11): “This finding suggests that USP28 undergoes dynamic turnover between 53BP1-bound and unbound pools, thereby enabling spatial separation of USP28 and 53BP1 after mitotic exit.”

9. I think Fig 3f shows that the dimer deficient variant Usp28-VLEE increases p53 levels in mitotic cells, despite a defect in 53bp1 interaction - this appears to contradict the authors' model.

Our model proposes that the 53BP1–USP28 complex forms during mitosis and regulates p53 stability after mitotic exit (Please see G1 release experiment in Fig. 2B). We don't think that the 53BP1–USP28 complex stabilizes p53 during mitosis. Since the dimer-deficient mutant (USP28-VLEE) can't form the complex during mitosis (Fig. 4G), we haven't tested the G1 release experiment, as we expect the mutant can't stabilize p53 in the G1 phase similar to other C-terminus binding deficient mutants (Fig. 5H).

10. The IF images shown in Fig. 5g,i,k and others should be quantified as done for some IF data (eg, Fig 2d).

Thank you for this helpful comment. We have now included quantifications for the IF images shown in Figures 5G, I, K, similar to the analysis presented in Figure 3D. These quantifications strengthen our conclusions by providing robust, quantitative support for the visual observations. See also Point 5 from Reviewer 2.

11. The last paragraph in the introduction is repeated twice.

We removed the duplicated paragraph.

Minor comments:

1. On page the authors write: "Although PLK1 is active only during mitosis...". This statement contradicts many studies showing S phase effects of Plk1 on DNA replication, etc. and should be rephrased.

We rephrased the text according to the reviewers suggestion:

Now: "Although PLK1 is primarily active during mitosis, this complex persists after cells are released from prolonged mitotic arrest into G1 phase (Fig. S6)"

2. What is the measure of confidence of the AlphaFold model that would support the following statement on page 6: "Structural predictions from AlphaFold with high-confidence modeling indicate that exon 19 forms an additional alpha helix in isoform 1, which may obstruct the interaction surface for 53BP1 (Fig. 1I; S2C)."

Thank you for pointing this out. We apologize for the original wording, which may have caused confusion. To clarify, AlphaFold's high-confidence structural predictions indicate that exon 19 forms an additional alpha helix in isoform 1, but the prediction does not directly address whether this helix obstructs the 53BP1 interaction surface. The suggestion that the helix may

interfere with 53BP1 binding is based on our experimental data rather than the AlphaFold model itself.

To improve clarity, we have rephrased the text accordingly and included the AlphaFold confidence scores related to the structural prediction (Figure S2C), emphasizing that the confidence pertains to the predicted fold, not the interaction interface.

The revised sentence now reads:

“Structural predictions from AlphaFold with high-confidence modeling indicate that exon 19 forms an additional alpha helix in isoform 1 (Fig. 2E; S2B, C). Our data suggest that this additional alpha helix obstructs the interaction surface for 53BP1 (Fig. 2D).”

3. On page 7 the authors write: “Since USP28 overexpression is toxic, we reasoned that all the observed mutations specifically impair p53 stabilization (Fig. 2B; S3A).

Overexpression of Usp28 is not generally toxic to tumor cells, as was shown by multiple previous studies. Perhaps it’s true in the context of mitotic stress, but this should be stated explicitly.

We clarified that the toxic effect of USP28 overexpression is related to observations in RPE1 cells, which is shown in the cited reference (PMID: 27371829).

Now: “Since USP28 overexpression is toxic in RPE1 cells ⁷, we reasoned that all the observed mutations specifically impair p53 stabilization (Fig. 2B; S3A).”

4. I don’t understand the meaning of the following passage on page 5 - “In contrast, USP28Δ had a less pronounced effect, and TP53BP1Δ had no effect on the sensitivity to DXR-induced DNA damage. The later observation could be explained by the independent role of 53BP1 in DNA repair.”

First, it’s surprising that 53bp1 loss doesn’t alter sensitivity to DXR. Second, this result rather suggests a 53bp1-independent role of Usp28 in sensitivity to DXR-induced DNA damage. This should be rephrased.

Initially, we hypothesized that 53BP1 might serve two roles at DNA damage sites: promoting DNA repair and activating p53. A failure in DNA repair could increase sensitivity to DXR, thereby masking any potential protective effect of 53BP1 deletion. We have rephrased the text to clarify this point. Additionally, we have incorporated the reviewer’s suggestion that USP28 may have a 53BP1-independent function in the DNA damage response.

Before: “In contrast, *USP28Δ* had a less pronounced effect, and *TP53BP1Δ* had no effect on the sensitivity to DXR-induced DNA damage. The later observation could be explained by the independent role of 53BP1 in DNA repair.”

Now: “The difference in effects between loss of *USP28* and *TP53BP1* could be explained by two potential models: (i) the absence of 53BP1-mediated DNA repair may increase sensitivity to DXR, which could counterbalance any protective effects normally provided by 53BP1; or (ii) the reduced sensitivity to DXR observed upon *USP28* loss may involve a 53BP1-independent function of *USP28* in the regulation of DNA damage responses.”

5. I also don't understand the following argument on page 7, “Notably, the expression levels of all tested mutants, except G903A and P953L, were 5- to 20-fold higher than in wildtype RPE1 cells (Fig. 2D; S3B, C), suggesting that the mutants fail to stabilize p53 following an extended mitotic duration even though the expression level is significantly increased.”

This assumes that p53 is the only *Usp28* effector, which is not true. Rephrasing this sentence can help.

We agree that the text was confusing. We rephrased the text according to the reviewers suggestion:

Before: “Notably, the expression levels of all tested mutants, except G903R and P953L, were 5- to 20-fold higher than in wildtype RPE1 cells (Fig. 3D; S3B, C), suggesting that the mutants fail to stabilize p53 following an extended mitotic duration even though the expression level is significantly increased. To assess p53 activation, we treated cells with the PLK4 inhibitor to prolong mitosis and measured stabilized p53 in the nucleus three days after the start of the treatment (Fig. 3D). We found that all mutant clones failed to sufficiently stabilize p53 after prolonged mitosis (Fig. 3D).”

Now: “The expression levels of all tested mutants, except G903R and P953L, were 5- to 20-fold higher than in wildtype RPE1 cells (Fig. 3D; S3B, C). To assess p53 activation, we treated cells with the PLK4 inhibitor to prolong mitosis and measured stabilized p53 in the nucleus three days after the start of the treatment (Fig. 3D). We found that all mutant clones, even when overexpressed, failed to sufficiently stabilize p53 after prolonged mitosis (Fig. 3D).”

6. On page 12, the authors write: “Using transgene expression, we found that the tumor suppressor function of *USP28* is mediated by the shorter isoform 2 of *USP28*...”.

As the authors do now directly analyze any effect on tumor suppression, this should be rephrased.

We rephrased the text according to the reviewers suggestion:

Before: “Using transgene expression, we found that the tumor suppressor function of *USP28* is mediated by the shorter isoform 2 of *USP28*, but not the longer isoform 1, which is frequently considered as canonical isoform.”

Now: “Using transgene expression, we found that p53 stabilization is mediated by the shorter isoform 2 of USP28, whereas the longer isoform 1 — often considered the canonical isoform — does not have this function

Reviewer #4 (Remarks to the Author):

Reviewer Response

REVIEWERS' COMMENTS

Reviewer #1 (Remarks to the Author):

All my comments have been addressed satisfactorily, and I am pleased with the thorough and thoughtful revisions. I would like to congratulate the Meitinger lab on this impressive first independent study.

Thank you very much for helpful comments and your kind words!

Reviewer #2 (Remarks to the Author):

The authors have addressed all the major critiques of the reviewers and I do not have any further comments.

We are grateful for your helpful comments during the revision process!

Reviewer #3 (Remarks to the Author):

In the revised version of the manuscript "Cancer-Associated USP28 Missense Mutations Disrupt 53BP1 Interaction and p53 Stabilization", Meitinger and co-workers have addressed most of the concerns raised in the original review. I don't have further questions. I believe it is an important and well-controlled study and should be interesting to many scientists in the field.

Thank you very much for helpful comments and your kind words!

Reviewer #4 (Remarks to the Author):

We also thank the trainee for reading our manuscript providing helpful input!